# GALOPA: Graph Transport Learning with Optimal Plan Alignment

**Yejiang Wang**[1,2]   **Yuhai Zhao**[1,2,†]   **Zhengkui Wang**[3]   **Ling Li**[1,2]

[1] School of Computer Science and Engineering, Northeastern University, China
[2] Key Laboratory of Intelligent Computing in Medical Image
of Ministry of Education, Northeastern University, China
[3] InfoComm Technology Cluster, Singapore Institute of Technology, Singapore
`wangyejiang@stumail.neu.edu.cn, zhaoyuhai@mail.neu.edu.cn,`
`zhengkui.wang@singaporetech.edu.sg, lilingneu@gmail.com`

## Abstract

Self-supervised learning on graphs aims to learn graph representations in an unsupervised manner. While graph contrastive learning (GCL - relying on graph augmentation for creating perturbation views of anchor graphs and maximizing/minimizing similarity for positive/negative pairs) is a popular self-supervised method, it faces challenges in finding label-invariant augmented graphs and determining the exact extent of similarity between sample pairs to be achieved. In this work, we propose an alternative self-supervised solution that (i) goes beyond the label invariance assumption without distinguishing between positive/negative samples, (ii) can calibrate the encoder for preserving not only the structural information inside the graph, but the matching information between different graphs, (iii) learns isometric embeddings that preserve the distance between graphs, a by-product of our objective. Motivated by optimal transport theory, this scheme relies on an observation that the optimal transport plans between node representations at the output space, which measure the matching probability between two distributions, should be consistent with the plans between the corresponding graphs at the input space. The experimental findings include: (i) The plan alignment strategy significantly outperforms the counterpart using the transport distance; (ii) The proposed model shows superior performance using only node attributes as calibration signals, without relying on edge information; (iii) Our model maintains robust results even under high perturbation rates; (iv) Extensive experiments on various benchmarks validate the effectiveness of the proposed method.

## 1 Introduction

Self-supervised graph learning involves learning representations of real-world graph data without the need for human supervision. Graph contrastive learning (GCL) [6, 16, 71] has been identified as one of the most successful graph self-supervised learning approaches, with its key components consisting of graph augmentation and contrastive learning. The former creates perturbation views of anchor graphs via various augmenting techniques, while the latter maximizes the similarity for two augmentations (positive pairs) of the same anchor and minimizes the similarity for those of two different anchors (negative pairs). The effectiveness of contrastive learning is dependent on the assumption that the augmented operations preserve the nature of data points and ensure that the augmented samples have consistent labels with the original ones.

---

† Corresponding author.

37th Conference on Neural Information Processing Systems (NeurIPS 2023).

However, graph data structures are discrete and their properties may vary significantly even with slight perturbations, which makes it much more challenging to design reasonable augmentations that guarantee the label-invariant assumption for graphs, in contrast to images or text. Additionally, the concept of "maximum similarity" in contrastive learning is difficult to measure since it is vague and lacks a clear indication of how much similarity should be maximized (or minimized) for a given pair of positive (or negative) views.

To address these challenges, an intuitive solution is to introduce the concept of distance from the input space, whereby the distance between the learned embeddings is forced to be equal to the distance between the corresponding input graphs. However, this requirement is challenging to achieve as the input objects (graphs) and output representations (vectors) are two distinct concepts, making it difficult to agree on their distance metrics. For instance, comparing the graph edit distance between graphs [15] and the Euclidean distance between vectors directly is inconceivable.

In response to the aforementioned challenges, in this paper, we propose a novel self-supervised learning method that seeks to align optimal transport plans [59] from graph space to node representation space, instead of transport distance [44]. This method is referred to as Graph Transport Learning (GTL), and it exploits the key concept in optimal transport (OT) theory, which aims to identify an optimal match (i.e., transport plan) between two data distributions (e.g., node sets of graphs) that minimizes the matching cost (i.e., transport distance). As shown in Figure 1, the transportation plan $\pi$ explicitly determines how to match particles from the source distribution $\mu$ to the target distribution $\nu$. Our approach involves several key steps. First, we obtain two graph views by augmenting a graph and generating node embeddings using a backbone model, such as Graph Neural Network (GNNs) [25]. Two optimal transport plans can be computed from the graph space and the representation space, respectively. To accurately capture the matching relationships in the original graph space, we enforce the backbone to learn representations that exhibit consistency with the optimal transport plans between the corresponding source graphs in the input space. This is

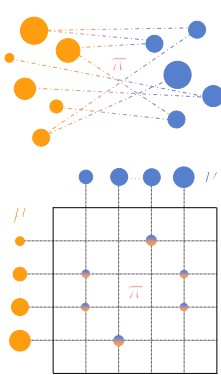

Figure 1: An illustration of discrete optimal transport plan.

achieved by minimizing the discrepancy between the two plans. Unlike distance-based approaches, the transport plan alignment shows several advantages: (i) *Direct comparability*: Plans with the same dimension can be compared directly, regardless of differences between the graph and representation spaces. The value of the transport plan is dimensionless, representing the joint probability of two particles; (ii) *Accurate match relationship*: For discrete objects, the transport plan retains more accurate matching relation between data compared to distance; (iii) *Label-variant*: Importantly, our self-supervised model does not require differentiating between positive and negative samples after graph augmentation. This is due to the availability of corrective information from the input space, which eliminates the need for a label-invariant assumption. Notably, our experimental findings in Section 7 demonstrate that our model maintains robust performance even under high perturbation rates, such as when 80% of both edges and node attributes are destroyed during graph augmentation. In summary, we make the following contributions:

- We propose a novel paradigm for self-supervised graph learning based on optimal plan alignment, GALOPA, which offers a distinct objective compared to contrastive learning methods. This approach eliminates the need for a label-invariant assumption.

- By constraining the discrepancy between the transport plans, we introduce a new loss to enable the sharing of the exact matching information from the graphs space to the representation space.

- Multiple comprehensive experiments, including distance v.s plan, node feature v.s edge, robustness test and comparison with state-of-the-art methods demonstrate remarkable performance of GALOPA.

## 2 Related Work

**Self-supervised Learning on Graphs.** Graph self-supervised learning has been a promising paradigm for learning useful representations of unlabeled graph data. The node embedding methods aim to learn a low-dimensional vector embedding of vertex preserving different kinds of order proximity via factorization [3], random walk [17] or deep learning [8, 61]. Recently, graph contrastive learning has achieved unprecedented success [6, 28, 55, 64], where the goal is to ensure representations have high similarity between positive views of a graph and high dissimilarity between negative views. A common way of contrastive objective is contrasting views at the node level. For the

representations (positive views) $\mathbf{h}_i^1$ and $\mathbf{h}_i^2$ of same node $i$ in two augmented graphs $\mathcal{G}_1$ and $\mathcal{G}_2$, the pairwise contrastive loss can be defined as

$$\mathcal{L}_{\text{contrast}} = \sum_i \log \left( \frac{e^{\mathcal{D}(\mathbf{h}_i^1, \mathbf{h}_i^2)/\tau}}{e^{\mathcal{D}(\mathbf{h}_i^1, \mathbf{h}_i^2)/\tau} + \sum_{k \neq i} e^{\mathcal{D}(\mathbf{h}_i^1, \mathbf{h}_k^2)/\tau}} \right) \tag{1}$$

where $\mathcal{D}$ is a discriminator that maps two views to an agreement (similarity) score, such as the inner product. $\tau$ denotes the temperature. Obviously, this type of loss strongly relies on the label invariance assumption. In other words, it needs to know (or assume) beforehand that the two views are positive/negative samples, which is challenging for discrete graph structures. For example, there exist some graphs, such as molecular graphs, whose labels are very sensitive to perturbation/corruption. Some studies have explored the label-invariant augmentation [29, 31, 74], but such augmentations require very careful design and adjustment, and sometimes limit the power of graph augmentation. More recently, [22] utilizes the graph edit distance to train graph discriminator to predict whether a graph is an original graph or a perturbed one. However, this model still requires positive and negative samples and relies on the assumption of label invariance.

**Optimal Transport.** Optimal transport (OT) [59] is a mathematical tool for aligning probability distributions has received much attention in the machine learning community in many applications, e.g., computer vision [4, 51], generative adversarial network [2, 7, 30], domain adaptation [18]. OT aims to derive a transport plan between a source and a target distribution, such that the transport cost is minimized. [34] proposes the Gromov-Wasserstein (GW) distance between metrics defined within each space rather than between samples across different spaces, which has been used as a distance between graphs in several applications [56, 60, 67]. However, these studies can only perform graph classification on datasets with multiple graphs and cannot be applied to network analysis, such as node representation classification, where only one network is available. In this work, we combine optimal transport problem with graph neural network to form a novel self-supervised learning paradigm that allows both graph and network learning. See Appendix A for more details.

## 3 Background

**Notations.** Let $\mathcal{G} = (\mathcal{V}, \boldsymbol{A}, \mu)$ be a graph of $n$ nodes where $\mathcal{V}$ denotes the set of nodes, $\boldsymbol{A} \in \{0,1\}^{n \times n}$ is the adjacency matrix. $\mu \in \mathbb{R}^n$ is the empirical distribution of nodes in $\mathcal{G}$. Generally, $\mu$ is a uniform discrete probability distribution. When the node label (or attribute) is available, we represent the node feature matrix as $\boldsymbol{X} \in \mathbb{R}^{n \times d}$, where $d$ denotes the dimension of node attributes. $\boldsymbol{X}^i$ represents the $i$-row of $\boldsymbol{X}$. Let $[\![n]\!] = \{1, \cdots, n\}$, $[\![n]\!]^2 = [\![n]\!] \times [\![n]\!]$ and $\langle \cdot, \cdot \rangle$ denotes the inner product for matrices. We denote $\otimes$ the tensor-matrix multiplication. $\mathbf{1}_n$ represents the vector with ones as all the $n$ elements. $|\cdot|$ denotes absolute value.

**Plan and Optimal Transport.** The optimal transport (OT) problem is pioneered by Monge [35] in order to seek the most cost-effective *transport plan* that transforms the mass of a pile of sand into another one. In particular, it studies how to find an optimal coupling or optimal plan $\pi$ for transforming the distribution $\mu$ to $\nu$ with minimum total transport cost (i.e., optimal transport distance), where the element of $\pi$ describes the probability of moving mass from one position to another. In this work, we mainly focus on the discrete case. Given two sets of features $\boldsymbol{X}_1 = \{\boldsymbol{X}_1^i\}_{i=1}^n$ and $\boldsymbol{X}_2 = \{\boldsymbol{X}_2^j\}_{j=1}^m$, where $n$ and $m$ are are the number of features, respectively. $\mu \in \mathbb{R}^n$ and $\nu \in \mathbb{R}^m$ are the probability distributions of the entities in the two sets, respectively. The formulation of the OT distance is

$$\mathcal{W}(\boldsymbol{X}_1, \boldsymbol{X}_2) = \min_{\pi \in \Pi(\mu, \nu)} \sum_{i \in [\![n]\!]} \sum_{j \in [\![m]\!]} c_{\mathcal{X}}(\boldsymbol{X}_1^i, \boldsymbol{X}_2^j) \cdot \pi_{ij} = \min_{\pi \in \Pi(\mu, \nu)} \langle \boldsymbol{K}(\boldsymbol{X}_1, \boldsymbol{X}_2), \pi \rangle \tag{2}$$

where

$$\Pi(\mu, \nu) = \{\pi \in \mathbb{R}^{n \times m} \mid \pi \mathbf{1}_m = \mu, \mathbf{1}_n \pi = \nu\} \tag{3}$$

denotes all the joint distributions $\pi$ with marginals $\mu$ and $\nu$. $\boldsymbol{K}(\boldsymbol{X}_1, \boldsymbol{X}_2)_{ij} = c_{\mathcal{X}}(\boldsymbol{X}_1^i, \boldsymbol{X}_2^j)$ is the cost (work) of moving $\boldsymbol{X}_1^i$ to $\boldsymbol{X}_2^j$, the cosine distance between $\boldsymbol{X}_1^i$ and $\boldsymbol{X}_2^i$ is a popular choice. The $\pi \in \mathbb{R}^{n \times m}$ is called as **transport plan**. This distance is also known as the Wasserstein distance.

**Optimal Transport for Graphs.** The Wasserstein problem requires the two distributions of point sets to lie in the same space. But for graphs, it is difficult to measure the cost between two nodes on

different graphs without node label (attribute). Even if the cost between nodes could be calculated, the Wasserstein distance cannot take the edge information into account. To compare distributions that are not necessarily in the same space, [34] defines Gromov-Wasserstein distance between two graphs without node label $\mathcal{G}_1 = (\boldsymbol{A}_1, \mu)$ and $\mathcal{G}_2 = (\boldsymbol{A}_2, \nu)$ as follow

$$\mathcal{W}_{\mathrm{GW}}(\mathcal{G}_1, \mathcal{G}_2) = \min_{\pi_{\mathcal{G}} \in \Pi(\mu,\nu)} \sum_{i,k \in [\![n]\!]^2} \sum_{j,l \in [\![m]\!]^2} c_{\mathcal{A}}(\boldsymbol{A}_1^{ik}, \boldsymbol{A}_2^{jl}) \cdot \pi_{ik}\pi_{jl} = \min_{\pi \in \Pi(\mu,\nu)} \langle \boldsymbol{L}(\boldsymbol{A}_1, \boldsymbol{A}_2) \otimes \pi, \pi \rangle$$
(4)

where $\boldsymbol{L}(\boldsymbol{A}_1, \boldsymbol{A}_2)$ is 4-D tensor and $\boldsymbol{L}(\boldsymbol{A}_1, \boldsymbol{A}_2)_{ijkl} = c_{\mathcal{A}}(\boldsymbol{A}_1^{ik}, \boldsymbol{A}_2^{jl})$. The cost function $c_{\mathcal{A}}$ is commonly defined as $c_{\mathcal{A}}(\boldsymbol{A}_1^{ik}, \boldsymbol{A}_2^{jl}) = |\boldsymbol{A}_1^{ik} - \boldsymbol{A}_2^{jl}|$.

Consider two graphs with node attributes $\mathcal{G}_1 = (\boldsymbol{A}_1, \boldsymbol{X}_1, \mu)$ and $\mathcal{G}_2 = (\boldsymbol{A}_2, \boldsymbol{X}_2, \nu)$ where $\boldsymbol{A}_1 \in \mathbb{R}^{n \times n}$ and $\boldsymbol{A}_2 \in \mathbb{R}^{m \times m}$ denote their adjacency matrices, $\boldsymbol{X}_1 \in \mathbb{R}^{n \times d}$ and $\boldsymbol{X}_2 \in \mathbb{R}^{m \times d}$ are feature matrices. The fused Gromov-Wasserstein distance [56] between graphs $\mathcal{G}_1$ and $\mathcal{G}_2$ can be defined as

$$\mathcal{W}_{\mathrm{FGW}}(\mathcal{G}_1, \mathcal{G}_2) = \min_{\pi_{\mathcal{G}} \in \Pi(\mu,\nu)} \sigma \sum_{ij} c_{\mathcal{X}}(\boldsymbol{X}_1^i, \boldsymbol{X}_2^j) \cdot \pi_{ij}^{\mathcal{G}} + (1-\sigma) \sum_{ijkl} c_{\mathcal{A}}(\boldsymbol{A}_1^{ik}, \boldsymbol{A}_2^{jl}) \cdot \pi_{ij}^{\mathcal{G}} \pi_{kl}^{\mathcal{G}} \quad (5)$$

which is equivalent to

$$\min_{\pi_{\mathcal{G}} \in \Pi(\mu,\nu)} \langle \sigma \boldsymbol{K}(\boldsymbol{X}_1, \boldsymbol{X}_2) + (1-\sigma)\boldsymbol{L}(\boldsymbol{A}_1, \boldsymbol{A}_2) \otimes \pi_{\mathcal{G}}, \ \pi_{\mathcal{G}} \rangle \quad (6)$$

where $\boldsymbol{K}(\boldsymbol{X}_1, \boldsymbol{X}_2)_{ij} = c_{\mathcal{X}}(\boldsymbol{X}_1^i, \boldsymbol{X}_2^j)$, $\boldsymbol{L}(\boldsymbol{A}_1, \boldsymbol{A}_2)_{ijkl} = c_{\mathcal{A}}(A_1^{ik}, A_2^{jl})$ and $\sigma \in [0; 1]$ denotes a trade-off parameter. Obviously, this definition is a fusion of Equations (2) and (4).

# 4 Graph Transport Alignment

As analyzed in the previous section, the recent graph contrastive techniques are deeply plagued by positive and negative sample generation, since graph properties could become completely different even with slight perturbations. To design a universal self-supervision scheme, we are motivated to calibrate the similarity between different representations in the output space using the matching signal between corresponding graphs from the input space. In particular, we first perturb the given graph to obtain two different graphs (e.g., a perturbed graph and the original one) and generate the node embeddings of the two graphs using the backbone model (e.g., GNNs). After computing the optimal transport plans for the graphs and the sets of node representations respectively (Sec-

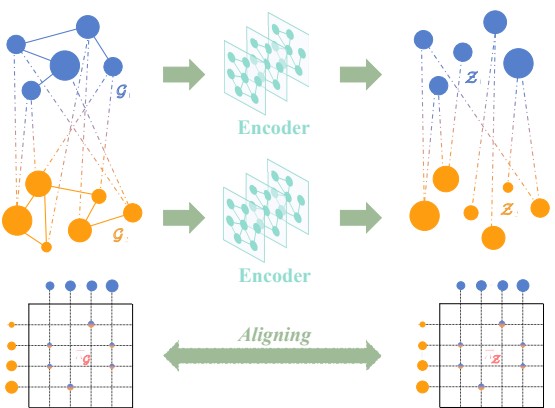

Figure 2: A framework of graph transport self-supervised learning.

tion 4.1), we take the discrepancy between the two plans as the loss to calibrate the backbone for obtaining representation with rich geometry awareness and interpretable correspondences (Section 4.2). We compare the proposed graph self-supervised learning paradigm with graph contrastive learning in Section 4.3. The framework can be found in Figure 2.

## 4.1 Optimal Transport Plan

In general, it is challenging to define two similarity metrics (e.g., distances), which can be directly compared, in two different spaces. This is especially the case for graphs and vectors, two distinct objects by nature. Fortunately, optimal transport theory offers a glimmer of hope, transportation plan, for such comparison. In this section, we present objectives that aim at finding the optimal plan for two graphs (or sets of vectors) to minimize the transport cost. For the graph, a natural idea is to jointly take into account both node attributes and explicit topology information (i.e., edges) in the transportation plan. In this work, we leverage [56] to present an objective function for calculating the optimal transport plan which integrates the edge structure and the feature information on nodes.

Specifically, the fused optimal transport plan $\pi_{\mathcal{G}}^* \in \mathbb{R}^{n \times m}$ between two node-attribute graphs $\mathcal{G}_1$ and $\mathcal{G}_2$ can be defined as

$$\pi_{\mathcal{G}}^* = \underset{\pi_{\mathcal{G}} \in \Pi(\mu, \nu)}{\text{argmin}} \ \langle \sigma \boldsymbol{K}(\boldsymbol{X}_1, \boldsymbol{X}_2) + (1 - \sigma) \boldsymbol{L}(\boldsymbol{A}_1, \boldsymbol{A}_2) \otimes \pi_{\mathcal{G}}, \ \pi_{\mathcal{G}} \rangle \tag{7}$$

with the fused Gromov-Wasserstein distance (6). By tuning the parameter $\sigma$ we can control the bias of the learned optimal plan between node attributes and edge structure. Intuitively we might think that combining more edge information could greatly benefit the expressiveness power of the model. However, we observe that the performance of the proposed method does not degrade significantly in the absence of edge information, and even increases rather than decreases on some datasets. It will be explained in detail later.

For the node representations of graph encoded by the backbone model (e.g., GNNs), we can either use Equation (2) directly to calculate the optimal plan $\pi_{\mathcal{Z}}^* \in \mathbb{R}^{n \times m}$ or set $\sigma = 1$ in Equation (7) as

$$\pi_{\mathcal{Z}}^*(\boldsymbol{Z}_1, \boldsymbol{Z}_2) = \underset{\pi_{\mathcal{Z}} \in \Pi(\mu, \nu)}{\text{argmin}} \ \langle \boldsymbol{J}(\boldsymbol{Z}_1, \boldsymbol{Z}_2), \ \pi_{\mathcal{Z}} \rangle \tag{8}$$

where $\boldsymbol{J}(\boldsymbol{Z}_1, \boldsymbol{Z}_2)_{ij} = c_{\mathcal{Z}}(\boldsymbol{Z}_1^i, \boldsymbol{Z}_2^j)$, $\boldsymbol{Z}_1$ and $\boldsymbol{Z}_2$ denote the node representations corresponding to $\mathcal{G}_1$ and $\mathcal{G}_2$, respectively.

## 4.2 Optimal Transport Alignment

The previous section establishes the foundation that comparing the similarity metrics defined on graph space and vector space can be reduced into comparing the two optimal transport plans from these spaces, each of which can be solved using Equations (7) or (2). This is a valid comparison because the optimal transport plan $\pi$ acts as a *probabilistic matching* of two distributions, while the two plan matrices have the same dimensions (i.e., $n \times m$).

Naturally, the encoder may succeed in obtaining a *good* representations for a graph if it learns the node embeddings that not only retain its structural information inside the graph, but also capture matching information with other graphs. This motivates us to force the encoder to preserve the matching relationship in the graph space by aligning the plan between the two graphs with the plan of their corresponding node representations. We define a match alignment loss by minimizing the discrepancy between the two transport plans as follows

$$\mathcal{L}_{\text{match}} = \Delta\left(\pi_{\mathcal{G}}^*, \ \pi_{\mathcal{Z}}^*(\boldsymbol{Z}_1, \boldsymbol{Z}_2)\right) \tag{9}$$

where the discrepancy function $\Delta(\cdot, \cdot)$ can be any commonly used metric, e.g., the Frobenius-norm $\|\cdot - \cdot\|_F$ or the divergence $D(\cdot\|\cdot)$.

In addition, to guide the encoder to learn a representation retaining structural information inside the graph, we also calibrate the cost matrix $\boldsymbol{J}(\boldsymbol{Z}_1, \boldsymbol{Z}_2)$, which implies the implicit structure relationships between nodes, in the representation space as follow

$$\mathcal{L}_{\text{(im)strc}} = \Delta\left(\sigma \boldsymbol{K}(\boldsymbol{X}_1, \boldsymbol{X}_2) + (1 - \sigma) \boldsymbol{L}(\boldsymbol{A}_1, \boldsymbol{A}_2) \otimes \pi_{\mathcal{G}}^*, \ \boldsymbol{J}(\boldsymbol{Z}_1, \boldsymbol{Z}_2)\right) \tag{10}$$

To understand the concept of '*implicit structure*' intuitively, here we consider the one-dimensional case. As shown in Figure 3, given a point set $\mathcal{P} = \{\boldsymbol{Z}_1^1, \cdots, \boldsymbol{Z}_1^n\}$, the location of each of its points is fixed. If the position of a point (yellow) $\boldsymbol{Z}_2^j$ is unknown but the transportation

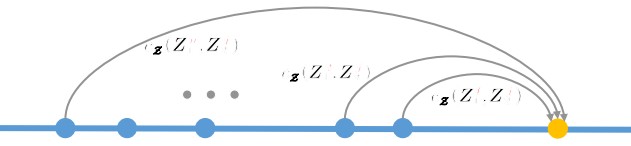

Figure 3: An illustration of implicit structure in one-dimension.

cost from this point to all points (blue) in the point set $\mathcal{P}$ is known, then the location of this point is determined with respect to the entire point set $\mathcal{P}$. The same holds true for another point $\boldsymbol{Z}_2^l$. Thus the relative position relationship between points $\boldsymbol{Z}_2^j$ and $\boldsymbol{Z}_2^l$ can be captured implicitly by the transport costs matrix $\boldsymbol{J}(\boldsymbol{Z}_1, \boldsymbol{Z}_2)$.

To this end, we define the overall graph transport alignment loss as

$$\mathcal{L}_{\text{GALOPA}} = \mathcal{L}_{\text{match}} + \rho \mathcal{L}_{\text{(im)strc}} \tag{11}$$

where $\rho$ is the trade-off parameter.

### 4.3 Compare with Graph Contrastive Learning

Although our objective function is different from the contrastive loss as shown in Equation (1), we find that the algorithmic philosophy of both is very similar. Here we analyze at the node level. Given a graph $\mathcal{G}$, the perturbation graphs $\mathcal{G}_1$ and $\mathcal{G}_2$ are obtained by augmenting $\mathcal{G}$. If the attribute and context of node $i$ in $\mathcal{G}$ are corrupted in a similar way and obtained two node views in $\mathcal{G}_1$ and $\mathcal{G}_2$. The cost between these two views is quite small (almost zero) and thus the optimal transport plan yields a high probability of matching between these two node views. With Equations (9) and (10), the model GALOPA calibrates the matching probability and cost of the corresponding node representations in the output space, which is actually making the representations of two similar nodes similar enough. And for two different nodes $i$ and $k$ of $\mathcal{G}$, the cost between their corresponding node views in graphs $\mathcal{G}_1$ and $\mathcal{G}_2$ is relatively larger. This leads to the opposite correction, i.e., making the representation of two dissimilar nodes sufficiently dissimilar.

From the contrastive learning perspective, the above process is not inconsistent with its goals. But there is a fundamental difference between graph transport alignment and graph contrastive learning: GALOPA directly utilizes calibration signal from the graph space. Precisely because of this signal, we do not have to distinguish between positive and negative samples, like *Maxwell's demon*.

### 4.4 Complexity

The time complexity of the model GALOPA is mainly influenced by the optimization process of Equations (7) and (8). To optimize Equation (7), which contains the fused Gromov-Wasserstein term, we utilize a conditional gradient (CG) solver [21]. This solver necessitates the computation of a gradient with a nearly cubic time complexity at each iteration concerning the size of the graph, i.e., the number of nodes. On the other hand, Equation (8) with the Wasserstein term can be optimized using the Sinkhorn-Knopp algorithm [12], which is highly time-efficient with a nearly square complexity.

## 5 Plan or Distance?

Thanks to the alignment of the plan and cost for the representation, as a byproduct, we find that the OT distance between the optimal node representations $\boldsymbol{Z}^*$ in Equation (11) is equal to the distance between its corresponding graphs. This is due to the two losses (10) and (9) constrain the optimal node representation to satisfy $\boldsymbol{J}(\boldsymbol{Z}_1^*, \boldsymbol{Z}_2^*) = \sigma \boldsymbol{K}(\boldsymbol{X}_1, \boldsymbol{X}_2) + (1-\sigma)\boldsymbol{L}(\boldsymbol{A}_1, \boldsymbol{A}_2) \otimes \pi_{\mathcal{G}}^*$ and $\pi_{\mathcal{Z}}^*(\boldsymbol{Z}_1^*, \boldsymbol{Z}_2^*) = \pi_{\mathcal{G}}^*$, respectively. This means that our losses can prompt the encoder to learn an isometric embedding that preserves the distance between graphs, which is one of the pursuits of the general representation model. Hence, we became interested in the question of who is more important, distance or plan? How would the model perform if we

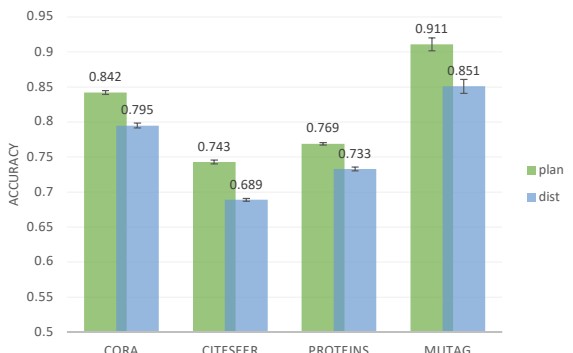

Figure 4: Plan versus distance. Comparing mean graph/node classification accuracy between transport alignment loss and distance loss on 4 datasets.

drop the alignment of the plan and cost but instead optimize the distance directly? In this section, we assess and rationalize the role of the plan for graph structure data in our self-supervised framework. To compare the possible performance gap between distance and plan, instead of directly optimizing the plan, we construct a new loss with distance as following

$$\mathcal{L}_{\text{dist}} = |\mathcal{W}_{\mathcal{G}}(\mathcal{G}_1, \mathcal{G}_2) - \mathcal{W}(\boldsymbol{Z}_1, \boldsymbol{Z}_2)| \tag{12}$$

where $|\cdot|$ denotes absolute value, and $\mathcal{W}_{\mathcal{G}}(\mathcal{G}_1, \mathcal{G}_2) = \min_{\pi \in \Pi(\boldsymbol{h}_1, \boldsymbol{h}_2)} \sigma \sum_{ij} c_{\mathcal{X}}(\boldsymbol{X}_1^i, \boldsymbol{X}_2^j) \cdot \pi_{ij}^{\mathcal{G}} + (1-\sigma) \sum_{ijkl} c_{\mathcal{A}}(A_1^{ik}, A_2^{jl}) \cdot \pi_{ik}^{\mathcal{G}} \pi_{jl}^{\mathcal{G}}$.

We evaluate the performance of using the pretraining representations on 2 social network datasets, CORA and CITESEER [25] for node classification, and 2 graph classification data PROTEINS and

MUTAG from TUDataset [36] for graph classification. See Section 8 for detailed experimental configurations. Figure 4 reports the averaged node/graph classification accuracy results over the node/graph-level datasets. The results suggest that the model using the plan as an objective significantly outperforms the counterpart models using the distance. Although the experimental result may lead to 'surprise', it demonstrates that the plan is closer to the essence than the distance, for discrete structured data. The optimal transport formulation Equation (5) contains both matching and implicit structural information. If only the final distance is retained instead of capturing the two types of information separately, the learned representation may fail to align properly with the input element. Because the optimal transport plan for the discrete OT problem is not unique in general and the optimal distance may correspond to several plans.

## 6    Node Attributes or Edges?

Among the information encoded in a graph, the structure and node attributes are two crucial elements for representation learning. The basic requirement of the encoder is to preserve the topology structures and capture the vertex feature of graphs. Thus a problem is encountered in our self-supervised learning paradigm: if we try to calibrate the representations learned by the backbone encoder, which is more important, the edge structure or the node attributes? In other words, if the calibration signal from the input space contains only node attribute information and completely ignores the explicit edge connectivity, will the performance of the proposed model deteriorate

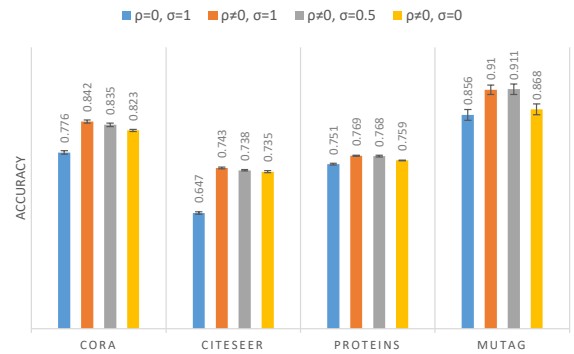

Figure 5: The mean graph/node classification accuracy on 4 datasets under different values of parameter $\sigma$.

significantly? The answer seems obvious—it should be. But the case seems to be different. Let's take a look at the experiment below.

As in the previous section, we performed the comparative experiments on four datasets CORA, CITESEER, PROTEINS, and MUTAG. Here we first consider the normal case of our loss function (i.e., $\rho \neq 0$). We set the value of the parameter $\sigma$ to adjust the bias of node attributes or edge connections for the plan in the graph space. If $\sigma = 1$, the model takes into account only node attributes in the transportation plans. When $\sigma = 0$, it integrates explicit edge information while completely ignoring the node feature. Figure 5 reports the results with different $\sigma$ and shows a surprising outcome: With only node attributes for the calculation of the plan, the model achieves outstanding performance on all the datasets, even optimal on some data. However, if we set $\rho = 0$ to remove the implicit structure constraint term $\mathcal{L}_{(\text{im})\text{strc}}$, the performance of the model suddenly deteriorates dramatically.

This verifies that the constraint $\mathcal{L}_{(\text{im})\text{strc}}$ is necessary and the implicit structural information it captures does calibrate the encoder even in the case of missing explicit edge connection. It therefore inspires self-supervised graph learning: it may not be able to tell that edge information does not contribute to correction for significant performance gains, but it is perfectly feasible to use only node attributes as a calibration supervisory signal for backbone model. This is a practical and valuable finding since the number of edges in a real-world graph dataset is much more than the number of nodes. The time and space complexity of the model can be reduced greatly if only the node information is used in the calculation of the correction signal while ignoring the edge connections.

## 7    Are the Transport Alignment Free from Positive/Negative Samples?

As shown in Equation (11), the proposed objective function does not distinguish whether the two different graphs/nodes are positive (negative) samples or not. Here, we want to show that the graph transport alignment strategy can be independent of the label invariance assumption. We conduct experiments below to see how different levels of perturbation affect the performance of GALOPA. When augmenting the graph, we fixed one of the augmentations as NoAug and the other augmentation

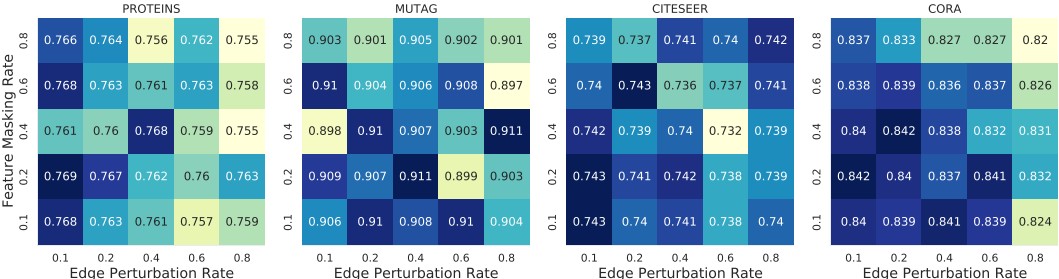

Figure 6: The mean graph/node classification accuracy when contrasting with different perturbation rates under 4 datasets. Fix one of the augmentations as NoAug and the other augmentation be the combination of edge perturbation and feature masking. Darker colors indicate better performance.

requires a hyper-parameter "*aug ratio*" that controls the portion of node attributes/edge that are selected for perturbing. Note that different augmentation strategies can be combined. Since the computation of the optimal transport plan in this paper involves node attributes and edge, we perform two augmentation policies, edge perturbation and feature masking, with different augmentation rates on four datasets (i.e., CORA, CITESEER, MUTAG, and PROTEINS) as shown in Figure 6.

From Figure 6 we find that the performance of our model does not change much even when the original graph is perturbed heavily. For example, even if eighty percent of the edges are removed while eighty percent of the node attributes are destroyed, the model's performance on the dataset CITESEER (0.742) is almost equal to the optimal result (0.743). Fluctuations of only zero point five to two percent are also observed on other data sets, such as one point three in PROTEINS data. Hence, it validates that the alignment of optimal transport between the source space and target space is indeed free from the label-invariant assumption.

## 8 Comparison with the State-of-the-art Methods

In this section, we compare our proposed self-supervised pre-training framework, GALOPA, with state-of-the-art methods in the settings of unsupervised learning on graph/node classification. More results can be found in the Appendix.

Table 1: Mean node classification accuracy (%) for supervised and unsupervised models. The highest performance of unsupervised models is highlighted in **boldface**. OOM indicates Out-Of-Memory.

| Model | CORA | CITESEER | PUBMED | WikiCS | Amz-Comp. | Amz-Photo | Coauthor-CS | Average |
|---|---|---|---|---|---|---|---|---|
| MLP | $47.92 \pm 0.41$ | $49.31 \pm 0.26$ | $69.14 \pm 0.34$ | $71.98 \pm 0.42$ | $73.81 \pm 0.21$ | $78.53 \pm 0.32$ | $90.37 \pm 0.19$ | $68.72 \pm 0.31$ |
| GCN | $81.54 \pm 0.68$ | $70.73 \pm 0.65$ | $79.16 \pm 0.25$ | $93.02 \pm 0.11$ | $86.51 \pm 0.54$ | $92.42 \pm 0.22$ | $93.03 \pm 0.31$ | $85.20 \pm 0.39$ |
| DEEPWALK | $70.72 \pm 0.63$ | $51.39 \pm 0.41$ | $73.27 \pm 0.86$ | $74.42 \pm 0.13$ | $85.68 \pm 0.07$ | $89.40 \pm 0.11$ | $84.61 \pm 0.22$ | $75.64 \pm 0.35$ |
| NODE2VEC | $71.08 \pm 0.91$ | $47.34 \pm 0.84$ | $66.23 \pm 0.95$ | $71.76 \pm 0.14$ | $84.41 \pm 0.14$ | $89.68 \pm 0.19$ | $85.16 \pm 0.04$ | $73.67 \pm 0.46$ |
| GAE | $71.49 \pm 0.41$ | $65.83 \pm 0.40$ | $72.23 \pm 0.71$ | $73.97 \pm 0.16$ | $85.27 \pm 0.19$ | $91.62 \pm 0.13$ | $90.01 \pm 0.71$ | $78.63 \pm 0.39$ |
| VGAE | $77.31 \pm 1.02$ | $67.41 \pm 0.24$ | $75.85 \pm 0.62$ | $75.56 \pm 0.28$ | $86.40 \pm 0.22$ | $92.16 \pm 0.12$ | $92.13 \pm 0.16$ | $80.97 \pm 0.38$ |
| DGI | $82.34 \pm 0.71$ | $71.83 \pm 0.54$ | $76.78 \pm 0.31$ | $75.37 \pm 0.13$ | $84.01 \pm 0.52$ | $91.62 \pm 0.42$ | $92.16 \pm 0.62$ | $82.02 \pm 0.46$ |
| GMI | $82.39 \pm 0.65$ | $71.72 \pm 0.15$ | $79.34 \pm 1.04$ | $74.87 \pm 0.13$ | $82.18 \pm 0.27$ | $90.68 \pm 0.18$ | OOM | – |
| MVGRL | $83.45 \pm 0.68$ | $73.28 \pm 0.48$ | $80.09 \pm 0.62$ | $77.51 \pm 0.06$ | $87.53 \pm 0.12$ | $91.74 \pm 0.08$ | $92.11 \pm 0.14$ | $83.67 \pm 0.31$ |
| GRACE | $81.92 \pm 0.89$ | $71.21 \pm 0.64$ | $80.54 \pm 0.36$ | $78.19 \pm 0.10$ | $86.35 \pm 0.44$ | $92.15 \pm 0.25$ | $92.91 \pm 0.20$ | $83.32 \pm 0.41$ |
| GCA | $82.38 \pm 0.47$ | $71.51 \pm 0.32$ | $80.89 \pm 0.28$ | $78.29 \pm 0.36$ | $87.88 \pm 0.26$ | $92.33 \pm 0.68$ | $92.64 \pm 0.34$ | $83.70 \pm 0.39$ |
| BGRL | $81.30 \pm 0.54$ | $72.06 \pm 0.63$ | $80.52 \pm 0.30$ | $76.13 \pm 0.18$ | $\mathbf{89.09 \pm 0.51}$ | $92.15 \pm 0.32$ | $92.33 \pm 0.39$ | $83.37 \pm 0.41$ |
| GALOPA | $\mathbf{84.21 \pm 0.30}$ | $\mathbf{74.34 \pm 0.18}$ | $\mathbf{84.57 \pm 0.34}$ | $\mathbf{81.23 \pm 0.19}$ | $88.65 \pm 0.11$ | $\mathbf{92.77 \pm 0.40}$ | $\mathbf{93.04 \pm 0.25}$ | $\mathbf{85.54 \pm 0.25}$ |

### 8.1 Experimental Setup

**Datasets.** We analyze the quality of representations learned by GALOPA on node and graph classification benchmarks. For node classification, we evaluate the performance of using the pretraining representations on 7 benchmark graph datasets, namely, CORA, CITESEER, PUBMED [25] and Wiki-CS, Amazon-Computers, Amazon-Photo, and Coauthor-CS [47]. For graph classification, we follow GRAPHCL [72] to perform evaluations on 6 graph classification data NCI1, PROTEINS, DD, MUTAG, COLLAB, and IMDB-B from TUDataset [36].

**Baselines.** For node-level tasks, we adopt three types of baselines: 1) *Supervised learning methods*, including MLP and GCN [25]; 2) *Graph embedding methods*, including DEEPWALK [42] and

NODE2VEC [17]; 3) *Graph contrastive learning methods*, including GAE, VGAE [24] , DGI [58], GMI [41], MVGRL [19], GRACE [77], GCA [78], and BGRL [55]. For graph-level task, we evaluate the performance of GALOPA in terms of the linear classification accuracy and compare it with 1) two *supervised learning methods*, including GCN [25] and GIN [69]; 2) seven *kernel-based methods*, including SP [5], GK [49], WL [50], WLPM [39], FGW [56], DGK [70], and MLG [26]; 3) three *unsupervised methods*, including NODE2VEC [17], SUB2VEC [1], GRAPH2VEC [37]; 4) five recent SOTA *self-supervised learning methods* based on contrastive learning, including INFOGRAPH [52], GRAPHCL [72], AD-GCL [53], JOAOV2 [73], RGCL [28] and SIMGRACE [66].

**Protocol.** We follow the standard evaluation protocol of previous state-of-the-art graph self-supervised learning approaches at the graph and node levels, respectively. Specifically, for graph classification, we report the mean 10-fold cross-validation accuracy after 5 runs followed by a linear SVM. The linear SVM is trained by applying cross-validation on training data folds and the best mean accuracy is reported. For node classification, we report the mean accuracy on the test set after 50 runs of training followed by a linear neural network model. For the graphs (nodes) datasets, we randomly split the data, where 80%/10%/10% (10%/10%/80%) of graphs (nodes) are selected for the training, validation, and test set, respectively.

**Implementation Details.** In the experiments, we use the Adam optimizer [23] with learning rate is tuned in $\{0.0001, 0.001, 0.01\}$. The optimization routine and the convergence analysis are summarized in Appendix B. We conduct the experiment with the trade-off parameter $\rho$ and $\sigma$, the parameter $C$ of SVM, batch size in the sets $\{10^{-3}, 10^{-2}, \ldots, 10^2, 10^3\}$, $\{0, 0.1, \ldots, 0.9, 1\}$, $\{10^{-3}, \ldots, 10^3\}$, $\{16, 64, 128, 256, 512\}$, respectively. To perform graph augmentation, we use 4 types of operations: Edge Perturbation, Feature Masking, Node Dropping, and Graph Sampling. Our model is implemented with Pytorch Geometric [13] and Deep Graph Library [63].

Table 2: Supervised and unsupervised representation learning classification accuracy (%) along with average accuracy of the algorithms on TU datasets. **Bold** indicates the best performance for unsupervised methods on each dataset. '–' means that the results are unavailable.

| Model | PROTEINS | DD | MUTAG | NCI1 | COLLAB | IMDB-B | Average |
|---|---|---|---|---|---|---|---|
| GCN | 74.92 ± 0.33 | 76.24 ± 0.14 | 85.63 ± 0.24 | 80.20 ± 0.14 | 79.01 ± 0.18 | 70.45 ± 0.37 | 77.74 ± 0.23 |
| GIN | 76.28 ± 0.28 | 78.91 ± 0.13 | 89.47 ± 0.16 | 82.75 ± 0.19 | 80.23 ± 0.19 | 73.70 ± 0.60 | 80.22 ± 0.25 |
| SP | 75.07 ± 0.54 | >1d | 85.25 ± 0.24 | 73.53 ± 0.16 | – | 55.62 ± 0.02 | – |
| GK | 71.67 ± 0.55 | 78.53 ± 0.03 | 81.71 ± 0.21 | 66.06 ± 0.12 | 71.81 ± 0.31 | 65.93 ± 0.10 | 72.61 ± 0.22 |
| WL | 72.92 ± 0.56 | 79.78 ± 0.36 | 80.76 ± 0.30 | 80.01 ± 0.50 | 69.30 ± 0.42 | 72.30 ± 0.44 | 75.84 ± 0.43 |
| WLPM | – | 78.79 ± 0.38 | 87.13 ± 0.42 | **86.32 ± 0.19** | – | – | – |
| FGW | 74.50 ± 0.23 | – | 88.34 ± 0.12 | 86.24 ± 0.31 | – | 62.97 ± 0.24 | – |
| DGK | 73.21 ± 0.61 | 74.79 ± 0.32 | 87.51 ± 0.65 | 79.98 ± 0.36 | 64.43 ± 0.48 | 67.09 ± 0.37 | 74.50 ± 0.46 |
| MLG | 41.23 ± 0.27 | >1d | 87.94 ± 0.16 | >1d | >1d | 66.67 ± 0.30 | – |
| NODE2VEC | 57.58 ± 0.36 | – | 72.62 ± 1.02 | 54.93 ± 0.16 | 56.12 ± 0.02 | 50.25 ± 0.09 | – |
| SUB2VEC | 53.06 ± 0.56 | 54.33 ± 0.24 | 61.17 ± 1.59 | 52.82 ± 0.15 | 55.26 ± 0.15 | 55.34 ± 0.15 | 55.33 ± 0.47 |
| GRAPH2VEC | 73.33 ± 0.21 | 79.32 ± 0.29 | 83.28 ± 0.93 | 73.21 ± 0.18 | 71.10 ± 0.54 | 71.16 ± 0.05 | 75.23 ± 0.36 |
| INFOGRAPH | 74.44 ± 0.31 | 72.85 ± 1.78 | 89.01 ± 1.13 | 76.20 ± 1.06 | 70.05 ± 1.13 | **73.03 ± 0.87** | 75.93 ± 1.04 |
| GRAPHCL | 74.39 ± 0.45 | 78.62 ± 0.40 | 86.80 ± 1.34 | 77.87 ± 0.41 | 71.36 ± 1.15 | 71.14 ± 0.44 | 76.69 ± 0.69 |
| AD-GCL | 73.28 ± 0.46 | 75.79 ± 0.87 | 88.74 ± 1.85 | 73.91 ± 0.77 | 72.02 ± 0.56 | 70.21 ± 0.68 | 75.65 ± 0.86 |
| JOAOV2 | 74.13 ± 0.51 | 77.32 ± 0.29 | 87.17 ± 1.09 | 78.40 ± 0.17 | 69.19 ± 0.16 | 70.37 ± 0.37 | 76.09 ± 0.43 |
| RGCL | 75.03 ± 0.43 | 78.86 ± 0.48 | 87.66 ± 1.01 | 78.14 ± 1.08 | 70.92 ± 0.65 | 71.85 ± 0.84 | 77.07 ± 0.74 |
| SIMGRACE | 75.23 ± 0.19 | 77.45 ± 1.03 | 89.27 ± 1.39 | 79.10 ± 0.25 | 71.37 ± 0.44 | 71.45 ± 0.29 | 77.31 ± 0.59 |
| GALOPA | **76.93 ± 0.18** | **83.87 ± 0.42** | **91.11 ± 1.27** | 77.86 ± 0.36 | **73.20 ± 0.37** | 70.72 ± 0.48 | **78.94 ± 0.51** |

## 8.2 Performance Comparison

**Performance under Node-level.** Table 1 reports the averaged results over the node-level datasets. Comparing the results in Table 1, we have the following major observations. The proposed method outperforms the state-of-the-art self-supervised models significantly and even exceeds the supervised models on several datasets. For example, on PUBMED, GALOPA achieves 84.57% accuracy, which is a 3.68% relative improvement over previous state-of-the-art unsupervised algorithms. When compared to supervised baselines, it outperforms strong supervised baselines: On CORA, CITESEER and PUBMED benchmarks we observe 2.67%, 3.61% and 5.41% relative improvement over GCN, respectively. On Coauthor-CS, the proposed unsupervised method shows competitive performance compared to the supervised models. Tabel 1 lists the average accuracy of 7 benchmark datasets, from which GALOPA achieves the best performance as well. For example, our proposed GALOPA

outperforms the unsupervised SOTA baseline SIMGRACE by 1.84% on average, and even outperforms supervised GCN by 0.34%. These results further validate that calibrating the backbone model by optimal transport alignment can produce expressive and generalizable representations.

**Performance under Graph-level.** In this section, we examine whether the proposed GALOPA performs better than state-of-the-art methods at graph-level datasets. The results of supervised learning baselines and unsupervised methods are reported in Table 2. The results shown in Table 2 suggest that GALOPA achieves state-of-the-art results with respect to unsupervised models. For example, on DD it achieves 83.87% accuracy, a 4.09% relative improvement over the previous state-of-the-art baselines. For kernel methods, our approach achieves better performance on most datasets. When compared to supervised baselines individually, our model outperforms GCN in 4 out of 6 datasets and outperforms GIN in 3 out of 6 datasets, e.g., a 1.64% relative improvement on GIN for the NCI1 dataset. Our approach outperforms the state-of-the-art graph contrastive learning approaches. For example, compared to SIMGRACE, which is one of the best SOTA methods, GALOPA has a relative improvement of 1.63% on average across all datasets. GALOPA outperforms GRAPHCL and INFOGRAPH with a relative improvement of 2.25% and 3.01% on average, respectively. To summarize, our newly proposed GALOPA for graph self-supervised has achieved SOTA performance.

## Discussion

**Conclusion.** In this paper, we investigated the self-supervised graph learning problem, addressing the challenges posed by label-invariant issues in contrasting graph learning. Unlike existing methods that adopt contrastive or distance-based regularization approaches, we propose a novel paradigm based on optimal transport for self-supervised graph learning. Our approach focuses on aligning optimal transport plans between the graph space and the representation space. By aligning the transport plans, our method enforces the backbone model to learn representations that precisely preserve the matching relationships in the original graph space. Our observations reveal several noteworthy findings: (i) The optimal transport plan serves as a more informative calibration signal for the encoder compared to the transport distance, capturing essential characteristics; (ii) It is feasible to utilize only node attributes as a correction signal for the backbone model, without relying on edge information; (iii) Our proposed graph self-supervised model eliminates the need to distinguish between positive and negative samples and overcomes the label-invariant assumption; Furthermore, extensive experiments conducted on multiple benchmark datasets demonstrate the state-of-the-art performance of our proposed framework in terms of generalizability and robustness.

**Limitations and Future Work.** Although the transport plan opens the door for direct communication between the input graph space and the output representation space, it also becomes a computational bottleneck for the model to some extent due to the limitation of optimal transport computation complexity. To reduce the time complexity, we can utilize the properties of the proposed model and/or the scaling optimal transport techniques that can reduce the time complexity from cubic to square or even to linear, we provide 4 ways to do this below: (i) Unlike general OT settings, where the two graphs are typically quite different and the matching relationship between them is completely unknown, the difference between the original and augmented graphs in GALOPA is quite small and the matching relations for subgraph components except with different part (i.e., complementary set of difference part) is known. This means that we can utilize the *matching prior* to reduce the computational cost. Hence, we can split the difference part with its neighborhood from the two graphs and compute the optimal transport plan only for that part. Since the percentage of that part is very small, it can greatly reduce the time complexity; (ii) According to the observation in Section 6, we can avoid the cubic complexity of optimizing GW by using only the node attributes for computing the optimal plan in graph space, while retaining similar performance with near-square time complexity; (iii) Alternatively, we can reduce the computational cost by utilizing sparsity [27] or graph partitioning [10, 67]. In particular, we can employ the most recent work on linear optimal transport [9, 38], which computes FGW term and/or Wasserstein term in linear time; (iv) We have the option to combine the aforementioned methods. For instance, by merging insights from the first point, a significant portion of subgraph pairs acquired via graph partitioning in the third point turns out to be identical. This realization can further pare down the complexity of partitioning methods.

As the main goal of this paper is to propose an alternative self-supervised graph learning paradigm beyond the label-invariant assumption that accurately links/communicates the input and output spaces, we leave the scalability issue as our future work.

## Acknowledgments and Disclosure of Funding

We thank Edouard Pauwels and Samuel Vaiter for their valuable work and for proving the convergence of the derivatives of Sinkhorn–Knopp. We also thank the anonymous reviewers for their constructive suggestions. This project was in part supported by the following projects: the National Natural Science Foundation of China (No.62032013, No.92267206); Singapore Institute of Technology Ignition Grant (No.R-IE2-A405-0001).

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
