# OpenReview forum: "GALOPA: Graph Transport Learning with Optimal Plan Alignment"
_NeurIPS.cc/2023/Conference — NeurIPS 2023 poster_

### Official Review · Reviewer_q42Z · 2023-06-28

**Soundness:** 3 good
**Presentation:** 4 excellent
**Contribution:** 3 good
**Rating:** 7
**Confidence:** 3

**Summary:**

The paper proposes a novel paradigm for self-supervised graph learning based on optimal plan alignment named GALOPA, which leverages optimal transport theory to align the optimal transport plans for graphs and node representations, resulting in an improvement in the quality of graph representations. Unlike existing methods, GALOPA does not require generating positive/negative sample pairs, simplifying the data requirements. GALOPA further introduces a new loss to enable the sharing of the exact matching information from graphs space to the representation space. The paper experimentally shows that GALOPA achieves state-of-the-art performance and robustness compared to previous methods on various benchmark datasets.

**Strengths:**

1.	The problem studied is an interesting and intuitive one that directly exploits and aligns the matching information common to both input and output spaces. This seems to combine the ideas of both graph contrastive learning and graph auto-encoders, while avoiding the limitations of both: the former needs to distinguish between positive and negative samples, while the latter needs to reconstruct the original signals (e.g. structure or node features), where the comparison takes place in the same space (graph space). The idea seems to be useful and extendable to other types of data (e.g. images or text) for self-supervised representation learning.
2.	The paper is well written and organized. I can easily understand the motivation and the proposed method. The illustrations can give readers a deeper understanding.
3.	The experimental design is good, and sections 5 and 6. They explore the rationality of the proposed methodology and dissect its validity. The interpretation of the experimental results is convincing. The results in section 8 verify that the algorithm is state of the art.


**Weaknesses:**

1.	The time complexity analysis may be better placed in the text than in the appendix.
2.	Some typos: L54 (generate->generating), L155 (node->nodes), L310 (require->requires).
3.	The authors should conduct Wilcoxon signed rank test to verify proposed method in Table 1-2.


**Questions:**

1、	The authors should provide more detailed information about the experimental setup, such as the size of the graph dataset used.
2、	The authors can provide a reasonable explanation for why using only node attributes yields the best performance in Section 6.
3、	Please review the paper and ensure that all the characters are in black color.
4、	In Figure 5, ‘ACCUTACY’ -> ‘ACCURACY’.


**Limitations:**

The authors have well discussed the limitations of their model and possible solutions.

---

> ### Author Rebuttal · Authors · 2023-08-09
>
> We sincerely thank the reviewer for describing our work as interesting and demonstrating superior performance and good design for the experiment. We respond to the reviewers’ questions below.
>
>
> > **Q1. The time complexity analysis may be better placed in the text than in the appendix.**
>
> Thanks for pointing this out. We will add time complexity analysis to the paper. In addition we have analyzed more ways to reduce time complexity in *GQ2* of the $\color{red}\text{global rebuttal}$.
>
>
> > **Q2. Some typos: L54 (generate->generating), L155 (node->nodes), L310 (require->requires). In Figure 5, ‘ACCUTACY’ -> ‘ACCURACY’.**
>
> Thanks for your valuable feedback on the typos. We will correct these errors and carefully check for any other errors that may exist in the paper.
>
> > **Q3. Conduct Wilcoxon signed rank test to verify proposed method in Table 1-2.**
>
> Upon the request of the reviewer, we performed Wilcoxon signed rank test on GALOPA and baseline on the node classification dataset and the graph classification dataset, respectively. Table VI and VII report the p-values for the Wilcoxon signed-ranks test for GALOPA at 0.05 significance level with node classification baselines and graph classification baselines, respectively. If the p-value is small, it can reject the idea that the difference is a due to chance and conclude that the population has a median distinct from the performance of baseline model.
>
> *Table VI.  The p-values for the Wilcoxon signed-ranks test on **node** classification datasets at 0.05 significance level.*
> |  GALOPA vs.  | p-value |
> | :--------: | :---: |
> |   BGRL   | 0.015 |
> | GCA | 0.007 |
> | GRACE | 0.007 |
> | MVGRL | 0.007 |
> | DGI | 0.007 |
> | VGAE | 0.007 |
> | GAE | 0.007 |
> | Node2Vec | 0.007 |
> | DeepWalk | 0.007 |
>
> *Table VII.  The p-values for the Wilcoxon signed-ranks test on **graph** classification datasets at 0.05 significance level.*
> |  GALOPA vs.  | p-value |
> | :--------: | :---: |
> |   SimGrace   | 0.078 |
> | RGCL | 0.078 |
> | Java2 | 0.046 |
> | AD-GCL | 0.015 |
> | GranpCL | 0.078 |
> | InfoGraph | 0.078 |
> | Graph2Vec | 0.031 |
> | Sub2Vec | 0.015 |
> | DGK | 0.031 |
> | WL | 0.078 |
> | GK | 0.015|
>
> As show in the table, GALOPA achieves superior performance against all the baselines.
>
>
>
> > **Q4. Provide more detailed information about the experimental setup, such as the size of the graph dataset used.**
>
> Thanking the reviewer q42Z for the suggestions, we add the description of the node classification and graph classification datasets used as follows
>
> *Table VIII. The statistical information of node classiﬁcation datasets.*
> |  Dataset  | Nodes | Edges | Classes | Feat. |
> | :--------- | ----: | -----: | ------: | ----: |
> |   Cora   | 2708 | 10556 | 7    | 1433 |
> |  CiteSeer  | 3327 | 9228  | 6    | 3703 |
> |  PubMed  | 19717 | 88651 | 3    | 500  |
> | WikiCS | 11701 | 216123 | 10 | 300 |
> |   Coauthor-CS   | 18333 | 327576 | 15   | 6805 |
> | Amz-Comp. | 13752 | 574418 | 10   | 767  |
> | Ama-Photo | 7650 | 287326 | 8    | 745  |
>
> *Table IX. The statistical information of graph classiﬁcation datasets.*
> |  Dataset  | Graphs | Avg. Nodes | Avg. Edges | Classes |
> | :--------- | ----: | -----: | ------: | ----: |
> |   PROTEINS   | 1113 | 39.06 | 72.82 | 2 |
> | DD | 1178 | 284.32 | 715.66 | 2 |
> | MUTAG | 188 | 17.93 | 19.79 | 2 |
> | NCI1 | 4110 | 29.87 | 32.30 | 2 |
> | COLLAB | 5000 | 74.49 | 2457.78 | 3 |
> | IMDB-B | 1000 | 19.77 | 96.53 | 2 |
>
> We'll add it to the paper.
>
> > **Q5. Provide a reasonable explanation for why using only node attributes yields the best performance in Section 6.**
>
> The main reasons why good performance can be achieved by utilizing only the node attributes in Section 6 are as follows: 1) The constraint $\mathcal{L_{(im)strc}}$ aims to correct the encoder so that the output node representations capture the implicit structure information. Thus, it also provides correction information to the encoder when explicit structural information (edges) is missing. 2) This implicit structural information may not only manifest the explicit structure, but in addition provide abundant auxiliary relation. Thus the best performance may be obtained even if only node attributes are used.
>
> > **Q6. Please review the paper and ensure that all the characters are in black color.**
>
> Thanks for pointing this out. We used red markers for some key concepts to make them easier for the reviewers to read, and we'll change those colors back to black afterwards.

---

> > ### Comment · Reviewer_q42Z · 2023-08-14
> >
> > Thanks for the authors' response. My concerns are well addressed.

---

> > > ### Author Response · Authors · 2023-08-17
> > > **Glad to hear that your concerns have been addressed.**
> > >
> > > We're glad to hear that we have addressed your concerns! Thanks for spending time on our submission, which makes our paper even stronger. **We will include these comparisons and results in the final version.**

---

### Official Review · Reviewer_bARL · 2023-07-05

**Soundness:** 3 good
**Presentation:** 3 good
**Contribution:** 2 fair
**Rating:** 5
**Confidence:** 4

**Summary:**

In this submission, the authors proposed a new self-supervised method for graph representation learning.
Unlike existing contrastive learning methods, the authors consider 1) the consistency between the optimal transport plan defined on the graph pairs and that defined on their node embeddings and 2) the consistency between the grounding cost defined on the graph pairs and the cost derived by the node embeddings.
Taking these two consistency terms as the objective function, the authors learn the representation model, achieving encouraging performance in node classification and graph classification tasks.
In addition, detailed analytic experiments are designed, providing valuable insights for graph self-supervised learning.

--- After rebuttal ---

Thanks for the authors' efforts in the rebuttal phase. Although the proposed method needs some modifications when dealing with heterophilic graphs, the methodology itself provides a new perspective of self-supervised learning for GNNs, and the experimental results are convincing. Therefore, my final score is "borderline accept".

**Strengths:**

1. The paper is easy to follow.

2. The analytic experiments are comprehensive. Some analytic results are interesting and can provide useful insights for the design of graph self-supervised learning methods.

3. The idea of leveraging the OT-based consistency between raw data and latent representation is interesting and differs from existing graph self-supervised learning methods.

**Weaknesses:**

1. The sigma in the fused GW term and the rho in the objective function are key hyperparameters, which may impact the performance of the proposed method significantly. However, the authors neither show the robustness of the method to the hyperparameters nor discuss the selection mechanism of the hyperparameters.

2. In my opinion, the computational complexity of the proposed method may be very high, which may limit the practical applications of the proposed method.

3. Some implementation details are missed, e.g., the algorithms to compute the fused GW distance between graphs and the OT distance between node embeddings. Additionally, it would be nice if the authors could provide an algorithmic scheme to show the whole self-supervised learning pipeline.

**Questions:**

1. When applying the proposed method, if the graphs are augmented randomly in different batches/epochs, we have to compute the fused GW distance between the augmented graphs iteratively. If my understanding is correct, the computational complexity of the method will be very high. How to solve/mitigate this problem?

2. For the node classification experiments in Table 1 and the analytic experiments in Figure 4, the authors seem to consider the homophilic graphs only. Could the authors test the proposed method on heterophilic graphs?

3. In Tables 1 and 2, the authors compared the proposed method with other graph contrastive learning methods. Are all the methods use the same backbone GNN? If the methods use different GNNs, how to ensure the fairness of the comparison?

**Limitations:**

The high computational cost may limit the application of the proposed method.

---

> ### Author Rebuttal · Authors · 2023-08-09
>
> We sincerely thank the reviewer for describing our work as interesting and for recognizing the valuable insights of our work for graph self-supervised learning. We respond to the reviewer’s concerns **below** and in the **global response above**.
>
> > **Q1. The sensitivity analysis of $\sigma$ and $\rho$, and discuss the selection mechanism.**
>
> For the selection of $\rho$ and $\sigma$, we search the optimal configuration for them from the set {$10^{-3}, 10^{-2}, \ldots, 10^2, 10^3$} and {$0, 0.1, 0.2, \ldots, 0.9, 1$}, which is also described in the Implementation Details section (Line 350) of the paper. At the request of the reviewer, we conducted robustness experiments on these two parameters. The following table shows the average node classification accuracy on the dataset Cora for different values of the parameter $\rho$ (vertical axis) and the parameter $\sigma$ (horizontal axis). For ease of viewing, we have bolded the values corresponding to non-zero $\rho$ and $\sigma$.
>
> *Table V. The sensitivity analysis of GALOPA on data Cora to the hyperparameters $\sigma$ and $\rho$.*
> |  $\rho$ vs. $\sigma$  | 0 | 0.3 | 0.5 | 0.8 | 1 |
> | :--------: | :---: | :--------: | :--------: | :--------: | :--------: |
> | **0** | 0.813±0.35 | 0.809±0.26 | 0.801±0.38 | 0.784±0.22 | 0.776±0.30 |
> | $10^{-3}$ | 0.816±0.21 | ***0.823±0.18*** | ***0.834±0.31*** | ***0.836±0.37*** | ***0.838±0.30*** |
> | $10^{-2}$ | 0.814±0.27 | ***0.828±0.45*** | ***0.831±0.36*** | ***0.840±0.32*** | ***0.838±0.21*** |
> | $10^{-1}$ | 0.818±0.34 | ***0.830±0.31*** | ***0.829±0.31*** | ***0.839±0.43*** | ***0.842±0.34*** |
> | $10^0$ | 0.823±0.24 | ***0.834±0.26*** | ***0.835±0.15*** | ***0.838±0.20*** | ***0.841±0.27*** |
> | $10^1$ | 0.819±0.29 | ***0.838±0.40*** | ***0.834±0.38*** | ***0.833±0.27*** | ***0.832±0.38*** |
> | $10^2$ | 0.821±0.18 | ***0.826±0.29*** | ***0.829±0.34*** | ***0.841±0.14*** | ***0.839±0.38*** |
> | $10^3$ | 0.820±0.33 | ***0.824±0.36*** | ***0.833±0.26*** | ***0.832±0.28*** | ***0.841±0.31*** |
>
> From the table we can see that when we **remove the implicit structure constraint** $\mathcal{L}_{(im)strc}$ ($\rho=0$), the performance of GALOPA drops dramatically if we do not use the explicit edge structure ($\sigma=1$) at the same time. Whereas, if we use the edge structure ($\sigma<1$) to a greater extent, i.e., the smaller the $\sigma$, the better performance of the algorithm. Additionally, we discuss the case where **only the node attributes are considered without using explicit edge structure** ($\sigma=1$). In this case if we add implicit structure constraints ($\rho \neq 0$) we can get superior performance.
>
> Combining these two cases, it can be concluded that **implicit structural constraint** $\mathcal{L}_{(im)strc}$ **do capture the internal structure of the graph**. Furthermore, we find that the algorithm is robust to the parameters $\rho$ and $\sigma$, which **fluctuate slightly for different values (>0)** of $\rho$ and $\sigma$.
>
>
> > **Q2. Provide the algorithms to compute the fused GW distance between graphs and the OT distance between node embeddings.**
>
> Reviewer can refer to the *GQ1* in the $\color{red}\text{global rebuttal}$.
>
>
> > **Q3. Reduce the complexity of  computing the FGW between the augmented graphs iteratively?**
>
> Reviewer can refer to the *GQ2* in the $\color{red}\text{global rebuttal}$.
>
> > **Q4. Could the authors test the proposed method on heterophilic graphs?**
>
> Thanks to the reviewers for the suggestions. However, our proposed algorithm mainly focuses on homophilic rather than heterophilic graph. If we want to process heterophilic graph well we may need to make some modifications to the algorithm, which may consume considerable time. Specifically, the implicit structural constraint $\mathcal{L_{(im)strc}}$ of the algorithm contains an assumption on homophilic graphs, i.e., the implicit structure captured by term $\mathcal{L_{(im)strc}}$ has the property that "**similar nodes tend to be neighborly**", which is contrary to the explicit edge connectivity assumption of heterophilic graph, i.e., **neighboring nodes are dissimilar**.
>
>
>
> > **Q5. Are all the methods use the same backbone GNN? If the methods use different GNNs, how to ensure the fairness of the comparison?**
>
> Yes, as mentioned by the reviewer, in order to ensure experimental fairness, we try to ensure that each algorithm uses backbone encoders with the same implementation (e.g., the number of layers and dimensions of the hidden layers).

---

> > ### Comment · Reviewer_bARL · 2023-08-11
> >
> > Thanks for the authors' reply. However, the rebuttal raises my concerns about the universality of the proposed method --- it seems that the proposed method cannot be applied to heterophilic graphs directly, and its usefulness across various GNN architectures is not investigated yet.

---

> > > ### Author Response · Authors · 2023-08-17
> > > **Response (1/2) to Reviewer bARL**
> > >
> > > Thank you for your valuable perspectives. We acknowledge any confusion that may have arisen from our previous response. Your concerns have been meticulously evaluated, and we aim to furnish a more comprehensive clarification. To better appreciate our work, we have encapsulated 3 crucial aspects, which are further elucidated through empirical evaluations.
> > >
> > > > 1) Universality of GALOPA: Our proposed GALOPA framework indeed possesses the flexibility to be employed on both homophilic and heterophilic graphs.  To adapt GALOPA for heterophilic graphs, we only need to replace the current backbone encoder with a suitable one tailored for heterophilic graph scenarios.
> > >
> > > According to [1], the heterophily restricts the learning ability of existing homophilic GNNs on general graph-structural data, resulting in significant performance degradation on heterophilic graphs. GALOPA is a flexible OT-based self-supervised framework.  The choice of a backbone for graph encoding in GALOPA is not rigidly tied to the proposed framework. This flexibility empowers users to select different backbones based on the specific context. For instance, transitioning GALOPA from homophilic to heterophilic graph settings involves a straightforward substitution of the current homophilic encoder with a suitable heterophilic encoder, serving as the backbone for GALOPA.
> > >
> > > To demonstrate this, we conducted a new set of experiments on 4 heterophilic graph data, i.e., *Chameleon, Wisconsin, Cornell, and Squirrel*. In these experiments, we compared GALOPA against state-of-the-art homophilic graph methods (BGRL) and heterophilic graph methods (SP-GCL [2]). For both BGRL and GALOPA, we assessed two scenarios by employing both traditional GNN encoders (HoGNN) used in the paper as well as specialized heterophilic GNN encoders (HeGNN) based on the structure proposed in [3]. We also retained SP-GCL's original encoder design as SP-GCL is specifically tailored for heterophilic graphs. The results are presented below.
> > >
> > > |  Alg.  | Wisconsin | Cornell | Squirrel | Chameleon |
> > > | :--------- | :---: | :----: | :-----: | :---: |
> > > |   BGRL(HoGNN)   | 0.523±0.27 | 0.561±0.34 | 0.462±0.31 | 0.634±0.51 |
> > > |  BGRL(HeGNN)  | 0.685±0.22 | 0.579±0.29 | 0.468±0.36 | 0.636±0.45 |
> > > | SP-GCL | 0.635±0.18 | 0.586±0.33 | **0.522±0.47** | 0.653±0.36 |
> > > | GALOPA(HoGNN) | 0.627±0.24 | 0.577±0.25 | 0.428±0.39 | 0.598±0.42 |
> > > | GALOPA(HeGNN) | **0.731±0.26** | **0.682±0.31** | 0.473±0.28 | **0.654±0.39** |
> > >
> > > The results demonstrate clear performance enhancements in GALOPA when transitioning the backbone from HoGNN to HeGNN across all heterophilic data. Notably, instances like the Wisconsin data exhibit a notable 16.6% enhancement (from 0.627 to 0.731), while Chameleon showcases a 9.4% uplift (from 0.598 to 0.654). Importantly, GALOPA consistently surpasses BGRL in performance when employing the same encoder. Additionally, in comparison to SP-GCL, a leading heterophilic graph solution, GALOPA outperforms it on three out of four datasets. This robust performance reinforces the efficacy of GALOPA on heterophilic graphs.
> > >
> > > > 2) Versatile Performance Across Graph Types: GALOPA demonstrates strong performance across both homophilic and heterophilic graph data, utilizing a single adaptable backbone.
> > >
> > > We demonstrate that if a graph encoder performs effectively on both homophilic and heterophilic graphs, the same holds true for GALOPA when utilizing this encoder as its backbone. The adaptability of HeGNN in encoding both homophilic and heterophilic graphs is evident [3]. To verify this, we evaluate the performance of GALOPA(HeGNN) on three homophilic graph data.
> > >
> > > |  Alg.  | Cora | CiteSeer | PubMed |
> > > | :--------- | :---: | :----: | :-----: |
> > > | GAE | 0.714±0.41 | 0.658±0.40 | 0.722±0.71 |
> > > | VGAE | 0.773±1.02 | 0.674±0.24 | 0.758±0.62 |
> > > | DGI | 0.823±0.71 | 0.718±0.54 | 0.767±0.31 |
> > > | GMI | 0.823±0.65 | 0.717±0.15 | 0.793±1.04 |
> > > | MVGRL | 0.834±0.68 | 0.732±0.48 | 0.800±0.62 |
> > > | GRACE | 0.819±0.89 | 0.712±0.64 | 0.805±0.36 |
> > > | GCA | 0.823+0.47 | 0.715±0.32 | 0.809±0.28 |
> > > | BGRL | 0.813+0.54 | 0.720±0.63 | 0.805±0.30 |
> > > | GALOPA(HoGNN) | **0.842±0.30** | 0.743±0.18 | **0.845±0.34** |
> > > | GALOPA(HeGNN) | 0.839±0.21 | **0.745±0.34** | 0.836±0.27 |
> > >
> > > The findings illustrate that GALOPA(HeGNN) exhibits comparable performance to GALOPA(HoGNN) on homophilic graphs while outperforming baseline methods. This can largely be attributed to its capacity to adeptly utilize the low-pass, high-pass, and identity channels within GNNs, effectively addressing the variations in both homophilic and heterophilic scenarios. These results further affirm GALOPA's capability to achieve strong performance across distinct graph types by utilizing a unified backbone.
> > >
> > > [1] Graph neural networks for graphs with heterophily: A survey. arXiv:2202.07082.
> > >
> > > [2] Can single-pass contrastive learning work for both homophilic and heterophilic graph?. arXiv:2211.10890.
> > >
> > > [3] Revisiting heterophily for graph neural networks. NeurIPS2022.

---

> > > ### Author Response · Authors · 2023-08-17
> > > **Response (2/2) to Reviewer bARL**
> > >
> > > > 3) Stability Across Various GNN Backbones: GALOPA exhibits consistent stability when employing different similar GNNs as its backbone.
> > >
> > > In compliance with the reviewer's request, we conducted an examination of the performance impact on GALOPA by employing diverse GNNs, specifically GCN (as used in the original paper) and SGC (with 1- or 2-hops, denoted as SGC-1 and SGC-2) [4] as encoders. The GCN structure employs a 2-layer design, while the SGC structure utilizes a 1-layer configuration by default. The hidden layer dimension for both models is set to 512.
> > >
> > > The results obtained from these experiments highlight the stability of GALOPA's performance when different GNNs are employed as its backbone. This consistency across diverse GNNs architectures underscores the robustness and versatility of our proposed approach. Also, we find that the performance of the model enhances with the increased expressive power of the backbone model. For example, GALOPA(SGC-2) outperforms GALOPA(SGC-1).
> > >
> > > |  Alg.  | Cora | CiteSeer | PubMed |
> > > | :--------- | :---: | :----: | :-----: |
> > > | MVGRL | 0.834±0.68 | 0.732±0.48 | 0.800±0.62 |
> > > | BGRL | 0.813+0.54 | 0.720±0.63 | 0.805±0.30 |
> > > | GALOPA(GCN) | **0.842±0.30** | **0.743±0.18** | 0.845±0.34 |
> > > | GALOPA(SGC-2) | 0.831±0.24 | 0.732±0.38 | **0.851±0.41** |
> > > | GALOPA(SGC-1) | 0.822±0.36 | 0.735±0.34 | 0.840±0.18 |
> > >
> > > [4] Simplifying graph convolutional networks. ICML2019.
> > >
> > > **Thank you for your valuable feedback. It will greatly aid in improving the quality of our manuscript. We will incorporate these comparisons and results into the final version.**

---

### Official Review · Reviewer_zpGL · 2023-07-07

**Soundness:** 3 good
**Presentation:** 4 excellent
**Contribution:** 3 good
**Rating:** 5
**Confidence:** 4

**Summary:**

This paper proposes a new paradigm for self-supervised graph learning  GALOPA based on optimal transport. It seeks to align optimal transport plans from graph space to node representation space instead of distance alignment in graph contrastive learning. The extensive experiments show that the optimal transport plan is more informative and overcomes the label-invariant assumption.

**Strengths:**

 - The problem to overcome the label-invariant assumption in augmentation-based graph self-supervised learning is interesting and significant. The idea to align the transport plan between the graph space and representation space is novel and interesting.
- The paper is well-organized and easy to follow. The introduction of related knowledge is very clear and helpful for understanding.
- The experiment can validate its conclusion.

**Weaknesses:**

- The connection and difference between the two parts of Loss is not clear
- The experiment is insufficient. It is better to add an ablation study on the two proposed losses to observe which part contributes much.
- This paper just uses the optimal transport method for self-supervised graph learning, which cannot be considered as a great innovation.

**Questions:**

- The authors can provide the algorithms concerning how to compute the transport plan （e.g. Sinkhorn-Knopp.)

- The authors use gradient descent to optimize the Eq(12) instead of Sinkhorn-Knopp. I wonder whether the accuracy of those optimized methods will affect the results.

**Limitations:**

Yes.

---

> ### Author Rebuttal · Authors · 2023-08-09
>
> We sincerely thank the reviewer for describing our work as interesting and significant. We respond to the reviewers’ questions below.
>
> > **Q1. The connection and difference between the two parts of Loss is not clear.**
>
> Thanks to the reviewer's suggestion, we describe the relationship between these two losses in more detail below. These two losses come from calibration of two aspects of the encoder: STRUCTURAL INFORMATION and MATCHING INFORMATION, both of which are expected to be captured in the output node representation of the encoder. $\mathcal{L_{match}}$ force the encoder to preserve the matching relationship between graphs in the graph space by minimizing the discrepancy between the two transport plans. $\mathcal{L_{(im)strc}}$ guide the encoder to learn a representation retaining structural information inside the graph by calibrating the cost matrix of node representations. Also from the optimal transmission perspective, these two losses focus on the optimal transport plan and the transportation cost, respectively, which together determine the optimal transport distance.
>
>
> > **Q2. Add an ablation study on the two proposed losses to observe which part contributes much.**
>
> At the request of the reviewer here we perform loss ablation experiments. We compare the performance of the algorithms when using Eq. (8) and Eq. (9) alone as losses below
>
> *Table IV. Ablation study on the losses $\mathcal{L_{match}}$ and $\mathcal{L_{(im)strc}}$.*
>
> |  Loss  | Cora | CiteSeer | PROTEINS | MUTAG |
> | :--------: | :---: | :--------: | :--------: | :--------: |
> | $\mathcal{L_{match}}+\mathcal{L_{(im)strc}}$ | **0.842±0.30** | **0.743±0.18** | **0.769±0.18** | **0.911±1.27** |
> |   $\mathcal{L}_{match}$   | 0.820±0.36 | 0.689±0.34 | 0.758±0.21 | 0.879±1.14 |
> | $\mathcal{L}_{(im)strc}$ | 0.812±0.25 | 0.684±0.24 | 0.762±0.23 | 0.869±1.21 |
>
> From the table we can see that using only one loss alone leads to performance degradation, which verifies that each loss is indispensable. In addition we find that the plan matching loss $\mathcal{L_{match}}$ gives relatively better performance on most datasets compared to the implicit structure loss $\mathcal{L_{(im)strc}}$, which also suggests that the former may contribute more.
>
>
>
> > **Q3. Provide the algorithms concerning how to compute the transport plan (e.g. Sinkhorn-Knopp).**
>
> We describe the optimization algorithms below.
>
> To solve the FGW problem of Eq. (6), we optimize the transport plan with the conditional gradient (CG) solver. The conditional gradient algorithm [1] consists in solving a linearization $\langle \mathbf{X}, \nabla_\pi \rangle$ at each iteration $r$. It can be solved by gradient descent with a direction $\mathbf{X}^{(r)} - \pi^{(r)}$, followed by a line search for the optimal step. The detail of the algorithm is summarized in Algorithm 1 in the Appendix.
>
> To solve the Wassersttein problem of Eq. (7), in the paper we use the Sinkhorn-Knopp algorithm [2] to iteratively approximate the optimal solution $\pi_z^*$. Specifically，Sinkhorn-Knopp algorithm add an additional entropy regularizer and perform a scheme of alternating Sinkhorn projections: $\pi^{(0)}=\exp(-\boldsymbol{J}(\boldsymbol{Z}_1, \boldsymbol{Z}_2)/\lambda)$ and $\pi^{(t+1)} = \mathcal{S} (\mathcal{T}(\pi^{(t)}))$, where $t$ denotes the number of iterations, $\lambda$ weights the regularization, $\mathcal{S}(\pi)=\pi \oslash(\mathbf{1} \mathbf{1}^{\top} \pi) \odot(\mathbf{1} \boldsymbol{b}^{\top})$ and $\mathcal{T}(\pi)=\pi \oslash(\pi \mathbf{1} \mathbf{1}^{\top}) \odot(\boldsymbol{a} \mathbf{1}^{\top})$, $\odot$ denotes the Hadamard product and $\oslash$ denotes element-wise division. As shown by [2], in the limit this scheme converges to a minimizer $\pi^{(t)} \stackrel{t \rightarrow \infty}{\longrightarrow} \pi^*$.
>
> [1] Revisiting frank-wolfe: Projection-free sparse convex optimization. ICML2013
>
> [2] Sinkhorn distances: Lightspeed computation of optimal transport. NeurIPS2013
>
>
>
> > **Q4.  The authors use gradient descent to optimize the Eq. (12) instead of Sinkhorn-Knopp. I wonder whether the accuracy of those optimized methods will affect the results.**
>
> Eq. (12) involves the Gromov-Wasserstein term rather than just the traditional Wasserstein term, and the Sinkhorn-Knopp algorithm is designed for Wassertein term and cannot be used to optimize the Gromov-Wasserstein term. We therefore use the conditional gradient (CG) solver, which often appears to optimize GW, to optimize Eq. (12). Thus, these two algorithms cannot be directly compared. Also as mentioned in the previous problem, for Eq. (7), which contains only Wasserstein terms, we optimize it with the Sinkhorn-Knopp algorithm.

---

> > ### Author Response · Authors · 2023-08-21
> > **Thanks for the review!**
> >
> > We extend our gratitude to the reviewer zpGL for acknowledging our work and providing us with valuable feedback.
> > **We will surely incorporate these comparisons, interpretations, and results in the final draft.**

---

### Official Review · Reviewer_YJ1t · 2023-07-11

**Soundness:** 2 fair
**Presentation:** 3 good
**Contribution:** 2 fair
**Rating:** 6
**Confidence:** 3

**Summary:**

In this paper, the authors study a new method "GALOPA" for self-supervised learning on graph.

For the two input views of graphs, they first compute the optimal transport plans w.r.t. fused GW distance between input graphs G1 and G2; for the corresponding outputs Z1=f(G1), Z2=f(Z2) of GNN f(), they compute the optimal transport plan w.r.t. a tunable cost J(Z1, Z2). They propose two losses to enforce the consistency between two plan matrices and two cost matrices.

They also propose some interesting findings through empirical experiments.

**Strengths:**

- originality:
    - This paper proposes a new type of loss based on **transport plan** for self-supervised learning on graph.

- quality:
    - The new framework they propose does work and can provide comparable performance to existing deep kernel methods.

- clarity:
    - They provide multiple illustrations to introduce the main ideas of different parts.

- significance:
    - This paper provides a new type of loss, which can be a good reference for self-supervised learning on graph.

**Weaknesses:**

- quality:
    - The analysis of computational complexity is not provided. From my perspective it can be an issue since the cost for doing OT is high, especially for GW distance.
    - The theoretical justifications for some claims are lacked. I also leave some questions below.
    - Some empirical results may be unconvincing due to the small size of datasets. E.g., the experiments in Section 5-7.

- clarity:
    - Some concepts are not clearly illustrated. See the questions below.

- significance:
    - The lack of theoretical results and insignificant improvement can make the paper less attractive to audience in this theoretical field.

**Questions:**

### Minor

1. The concept of “maximum similarity” in Line 35 is not well explained.
2. Could you further explain why it "is challenging for discrete graph structures to know (or assume) beforehand that the two views are positive/negative samples" in Line 92-93. Solving it seems an important contribution of the paper, while I don't get it here.
3. It is better to directly introduce the concept of fused GW distance in Section 3, and therefore better convey the idea in Section 4.1.
4. Some strong claims (in Section 5-7) are made based on experiments on small datasets. Can the authors add some medium to large datasets, e.g. from Open Graph Benchmark (OGB), to the experiments?


### Major

1. The implementation of "GALOPA" is not clearly introduced. For me, the authors propose two new losses, while the backbone can be any models for obtaining graph node embedding. I don't understand why in the experiments we contrast some backbone models (MLP, GCN) to "GALOPA".
2. As mentioned, the analysis of computational complexity is not provided. Please also compare it to common baseline methods.
3. The loss $L_{match}$ needs more discussions. Specifically, I feel it is non-trivial to discuss the differentiability of $\pi_z^*(Z_1, Z_2)$. It is hard to obtain the explicit form of the gradient of it w.r.t. $Z_i$, and I'm not sure whether the auto differentiation in common DL package can ensure the gradient in use will exactly be the real gradient of $\pi_z^*(Z_1, Z_2)$, considering the complex procedure of obtaining the OT plan.
4. Can you further explain the claim in Line 236 that "the OT distance between the optimal node representations Z∗ in Equation (10) is **equal** to the distance between its corresponding graphs".
    - Eqn. (10) is composed of two losses, so it is better to specify more clearly the "OT distance between the optimal node representations".
    - Can you show the derivation why the "OT distance" is "equal to the distance between its corresponding graphs"?

---

> ### Author Rebuttal · Authors · 2023-08-09
>
> We sincerely thank the reviewer for constructive feedback and for describing our work as good reference for self-supervised learning and demonstrating interesting findings in the experiments. We respond to the reviewer’s concerns **below** and in the **global response above**.
>
> > **Q1. Large scale data in Section 5-7?**
>
> We performed the experiments in Sections 5-7 on the dataset ogb-arxiv, which contains 169,343 nodes with 1,166,243 edges. We first partition the graph and compute the plan between subgraphs. We **perform self-supervised pre-training** with all training data and **supervised fine-tuning with 10% of them** then evaluate on the test sets, which is repeated for 10 times.
>
> In **Section 5**, we compare the plan and distance. Table I records the node classification accuracies on the ogb-arxiv when pre-training using the Eqs. (10) and (11) as losses.
>
> *Table I. Plan versus distance on large-scale ogb-arxiv.*
>
> |  Algo.  | Test |
> | :--------: | :--------: |
> |   NoPretrain   | 0.512±0.31 |
> | Plan | **0.544±0.18** |
> | Dist | 0.527±0.26 |
>
> ‘NoPretrain’ refers to direct fine-tuning without pre-training. We can find that using the plan as losses outperforms the counterpart using the distance, which is consistent with the conclusions in the paper.
>
> In **Section 6**, we compare node attributes and edges. If $\sigma=1$, the model takes into account only node attributes. When $\sigma=0$, it integrates only edge. We set $\rho = 0$ (or $\rho \neq 0$) to remove (or add) the implicit structure term $\mathcal{L}_{(im)struc}$.
>
> *Table II. Results on ogb-arxiv under different values of parameters.*
>
> |  Para  | Test |
> | :--------: | :--------: |
> |   $\rho=0, \sigma=1$   | 0.523±0.24 |
> | $\rho \neq 0, \sigma=1$ | 0.542±0.37 |
> | $\rho \neq 0, \sigma=0.5$ | 0.544±0.34 |
> | $\rho \neq 0, \sigma=0$ | 0.540±0.25 |
>
> From the table we can see similar conclusions to the paper.
>
> In **Section 7**, we test the robusness of GALOPA. Table III records the performance  when both feature masking (vertical axis) and edge perturbations (horizontal axis) are used.
>
> *Table III. Results on ogb-arxiv with diffrent perturbation rate.*
>
> |  Aug Rate  | 0.1 | 0.2 | 0.4 | 0.6 | 0.8 |
> | :--------: | :---: | :--------: | :--------: | :--------: | :--------: |
> |   0.1   | 0.542±0.23 | 0.543±0.36 | 0.540±0.32 | 0.538±0.31 | 0.532±0.26 |
> | 0.2 | 0.543±0.30 | **0.544±0.27** | 0.544±0.18 | 0.543±0.35 | 0.538±0.45 |
> | 0.4 | 0.539±0.16 | 0.541±0.33 | **0.544±0.29** | 0.542±0.17 | 0.540±0.39 |
> | 0.6 | 0.534±0.38 | 0.539±0.19 | 0.543±0.26 | 0.542±0.32 | 0.539±0.19 |
> | 0.8 | 0.531±0.23 | 0.535±0.25 | 0.539±0.19 | 0.537±0.36 | 0.537±0.41 |
>
> We can find that our model is robust on ogb-arxiv.
>
>
> > **Q2. Backbone (MLP, GCN) on experiment?**
>
> In the experiments, our aim is not to contrast GALOPA with different backbones. Instead, we aim to assess the performance gap between GALOPA (with GNNs backbone) and established supervised methods (e.g., MLP algorithm [1] and GCN algorithm [2]). This comparison intends to reveal the performance difference between unsupervised GALOPA and the supervised approach.
>
> [1] Graph attention networks. STAT2017
>
> [2] Semi-supervised classiﬁcation with graph convolutional networks. arXiv2016
>
>
>
> > **Q3.  Analysis of computational complexity.**
>
> Reviewer can refer to the *GQ2* in the $\color{red}\text{global rebuttal}$.
>
>
> > **Q4.  Differentiability of** $\pi_Z^*(Z_1, Z_2)$.
>
> It is difficult to derive the optimal closed-form solution for the Eq.(7) and the differentiation of $\pi_z^*(Z_1, Z_2)$ with respect to $Z_i$ directly. To solve the problem, we use the Sinkhorn-Knopp algorithm [1] to iteratively approximate the optimal solution $\pi_Z^*$, where the derivatives in each iteration step are solvable. Specifically, Sinkhorn-Knopp algorithm add an additional entropy regularizer and perform a scheme of alternating Sinkhorn projections: $\pi^{(0)}=\exp(-\boldsymbol{J}(\boldsymbol{Z}_1, \boldsymbol{Z}_2)/\lambda)$ and $\pi^{(t+1)} = \mathcal{S} (\mathcal{T}(\pi^{(t)}))$, where $t$ denotes the number of iterations, $\lambda$ weights the regularization, $\mathcal{S}(\pi)=\pi \oslash(\mathbf{1} \mathbf{1}^{\top} \pi) \odot(\mathbf{1} \boldsymbol{b}^{\top})$ and $\mathcal{T}(\pi)=\pi \oslash(\pi \mathbf{1} \mathbf{1}^{\top}) \odot(\boldsymbol{a} \mathbf{1}^{\top})$, $\odot$ denotes the Hadamard product and $\oslash$ denotes element-wise division. As shown by [1], in the limit this scheme converges to a minimizer $\pi^{(t)} \stackrel{t \rightarrow \infty}{\longrightarrow} \pi^*$. Hence, the differential $\partial\pi^{(t)}/\partial Z_i$ can be solved by the chain rule.
>
> [1] Sinkhorn distances: Lightspeed computation of optimal transport. NeurIPS2013
>
> > **Q5. Can you explain the claim that "the OT distance between the optimal node representations** $\boldsymbol{Z}^*$ **in Eq. (10) is equal to the distance between its corresponding graphs" (Line 236)?**
>
> We apologize for the distress caused by our lack of explanation here. Eq. (10) is obtained by adding the two loss (Eq. 9 and 8), which constrain the optimal node representation $\boldsymbol{Z_i}^*$ to satisfy  $\boldsymbol{J}(\boldsymbol{Z_1}^*, \boldsymbol{Z_2}^*)=\sigma \boldsymbol{K}(\boldsymbol{X_1}, \boldsymbol{X_2})+(1-\sigma) \boldsymbol{L}(\boldsymbol{A_1}, \boldsymbol{A_2}) \otimes \pi_\mathcal{G}^*$ and $\pi_\mathcal{Z}^*(\boldsymbol{Z_1}^*, \boldsymbol{Z_2}^*) = \pi_\mathcal{G}^*$, respectively. Hence, the OT distance between the optimal node representations $\mathcal{W}(\boldsymbol{Z_1}^*, \boldsymbol{Z_2}^*)=\langle\boldsymbol{J}(\boldsymbol{Z_1}^*, \boldsymbol{Z_2}^*), \pi_\mathcal{Z}^*(\boldsymbol{Z_1}^*, \boldsymbol{Z_2}^*)\rangle$ is equal to the distance between its corresponding graphs $\mathcal{W}_G(\mathcal{G}_1, \mathcal{G}_2)=\langle\sigma \boldsymbol{K}(\boldsymbol{X_1}, \boldsymbol{X_2})+(1-\sigma) \boldsymbol{L}(\boldsymbol{A_1}, \boldsymbol{A_2}) \otimes \pi^*_G,\ \pi^*_G\rangle$.
>
> We place the remaining 3 questions Q6-Q8 below.

---

> ### Author Response · Authors · 2023-08-10
> **We place the remaining 3 questions Q6-Q8 below.**
>
> > **Q6. The explantation of “maximum similarity” in Line 35?**
>
> "Maximize similarity" in line 35 refers to maximizing the agreement score between two graphs. The agreement is usually measured by similarity score, such as inner product, between two representations. Given training graphs, graph contrastive learning aims to learn graph encoder such that representations of similar graphs (i.e., original and augmented graph) agree with each other.
>
> > **Q7. Why it "is challenging for discrete graph structures to know (or assume) beforehand that the two views are positive/negative samples" in Line 92-93?**
>
> There exist some graphs, such as molecular graph, whose label are very sensitive to perturbation/corruption. In other words, even with a very slight perturbation, the property (label) of the perturbed graph may change with respect to the original graph. Thus, it is difficult to determine whether it is a positive or negative sample after the perturbation. For example, in a molecular activity classification task, the activity (i.e., label) of a molecular graph may come from a certain functional group. A slight perturbation of this functional group can result in molecular inactivation, i.e., the label of the augmented molecular graph is changed relative to the original molecular graph. For such cases, graph contrastive learning will learn similar representation for semantically dissimilar graphs.
>
> > **Q8. It is better to directly introduce the concept of fused GW distance in Section 3.**
>
> Thanks to the reviewer's suggestion, we will include a description of the FGW distance in Section 3.

---

> ### Comment · Reviewer_YJ1t · 2023-08-16
>
> Thanks for the clarification in the author response. I still have two remaining concerns as follows.
>
> For the computational complexity, the author proposed some resolutions to reduce the cost.
> - Is it always the case that we can know the so-called matching prior of the two graphs $G_1, G_2$ involved in the computation? I can understand we may know that for augmented data samples, while what if $G_1, G_2$ are just two samples under different labels and are quite different?
> - The fast algorithms must have some cost, while I'm not sure how the cost of the fast algorithms would influence the proposed model/loss.
> - It might be more convincing if the new experiments on ogbn-arxiv can be timed for both the proposed method and related baselines.
>
> For the usage of $L_{match}$,
> - Do you in practice use the gradient from $\partial \pi^{(T)} / \partial Z_i$ to update the parameters in GNN $f()$?
> - If yes, even we know $\pi^{(T)} \to \pi^*$, how would you show the gradient $\partial \pi^{(T)} / \partial Z_i \to \partial \pi^* / \partial Z_i$?

---

> > ### Author Response · Authors · 2023-08-20
> > **Response (1/3) to Reviewer YJ1t**
> >
> > We're glad to hear that we have addressed some of the reviewer's concerns, and we'll respond to the rest of the reviewer's questions below.
> >
> > > **Q9-1. Is it always the case that we can know the matching prior of the two graphs $G_1$, $G_2$ involved in the computation?  What if $G_1$, $G_2$ are just two samples are different?**
> >
> > We greatly appreciate the reviewer's consideration of the matching prior aspect in our proposed methodology. In our previous response, we suggested several ways to reduce the computational expense. If $G_1$ and $G_2$ are distinct samples with diverse labels, the other solutions, which do not rely on prior knowledge of the relationship between $G_1$ and $G_2$, can still work effectively.
> >
> > Specifically, for dataset comprising a sole graph (e.g., citation network, social network), we need to use augmentation strategies to obtain multiple samples. The matching prior between these graphs is known in this scenario.
> >
> > For datasets encompassing multiple graphs, our model adapts well to scenarios involving augmented or entirely distinct graphs. Let's delve into the scenario of employing two entirely distinct graphs:
> >
> > 1) In cases where the datasets contain small to modest-sized graphs (e.g., with an average of fewer than 1000 nodes per graph), the computational complexity associated with matching is negligible. Here, the original model can be effectively utilized.
> >
> > 2) Furthermore, for datasets that comprise large graphs, alternative strategies are available to mitigate computational complexity. Techniques such as the utilization of only node attributes (point 2 in **GQ2**), graph partitioning, or the linear optimal transport algorithm (point 3 in **GQ2**), offer avenues for reducing time complexity. It's crucial to highlight that these approaches don't necessitate prior matching information of the two graphs, rendering them versatile options.

---

> > ### Author Response · Authors · 2023-08-20
> > **Response (2/3) to Reviewer YJ1t**
> >
> > > **Q9-2. I'm not sure how the cost of the fast algorithms would influence the proposed model/loss?**
> >
> >
> > As rightly pointed out by the reviewer, fast algorithms do indeed come with their associated trade-offs. Efficiency gains may entail some compromise in precision. In light of this observation, our subsequent comparison experiments (Tables (1)-(5)) illustrate that the performance trade-off resulting from these fast algorithms is minor when compared to the significant time savings they offer. This holds particularly true for medium to large datasets. Thus, the cost incurred by the use of fast algorithms is judiciously balanced against the benefits they provide.
> >
> > To address the reviewer's concerns and provide a more comprehensive evaluation, we've devised a variant algorithm known as GALOPA(linear). This variant focuses on computing the transport plan solely based on the node attributes and employs the linear Sinkhorn algorithm [1] to optimize in both the graph space and representation space. The original version of our model retains the name GALOPA(cube).
> >
> > We have meticulously conducted experiments, maintaining the experimental settings outlined in the paper, across node and graph classification datasets. The performance outcomes are diligently recorded, and the results are presented in the subsequent table for your reference.
> >
> > From the results on Tables (1)-(2) we can observe that the variant GALOPA(linear) exhibits comparable performance with the GALOPA(cube), especially on median/large graphs such as the Ama-Photo (with 7,650 nodes), PubMed (19,717 nodes), Coauthor-CS (18,333 nodes), and Amz-Comp. (13,752 nodes). In some cases, GALOPA(linear) performs better than GALOPA(cube) because the neural networks may get stuck at local optima resulting in a slight difference in performance, which is a side note to the good performance of GALOPA(linear).
> >
> > *Table (1): Node classiﬁcation accuracy (%) for GALOPA(cube) and GALOPA(linear).*
> > |  Models  | Cora | CiteSeer | PubMed | WiKiCS | Amp-Comp. | Amp-Photo | Coauthor-CS |
> > | :--------: | :---: | :--------: | :--------: | :--------: | ---------- | ---------- | ---------- |
> > |  **GALOPA(cube)**  | 84.21±0.30 | 74.34±0.18 | 84.57±0.34 | 81.23±0.19 | 88.65±0.11 | 92.77±0.40 | 93.04±0.25 |
> > |   **GALOPA(linear)**   | 82.73±0.29 | 72.12±0.35 | 84.39±0.19 | 81.15±0.39 | 88.49±0.17 | 92.82±0.27 | 92.76±0.22 |
> >
> > *Table (2): Graph classiﬁcation accuracy (%) for GALOPA(cube) and GALOPA(linear).*
> > |  Models  | PROTEINS | DD | MUTAG | NCI1 | COLLAB | IMDB-B |
> > | :--------: | :---: | :--------: | :--------: | :--------: | ---------- | ---------- |
> > |  **GALOPA(cube)**  | 76.93±0.18 | 83.87±0.42 | 91.11±1.27 | 77.86±0.36 | 73.20±0.37 | 70.72±0.48 |
> > |   **GALOPA(linear)**   | 76.77±0.32 | 82.39±0.45 | 90.88±1.29 | 76.59±0.24 | 73.33±0.41 | 70.71±0.39 |
> >
> > Additionally, we count the average elapsed time per epoch for training these two models on all datasets.
> > Note that all the experiments are conducted and runtimes are recorded on the *same hardware environment* as stated in the paper. The results are shown in the table below. These tables underscore the substantial reduction in time consumption associated with GALOPA(linear) compared to GALOPA(cube), especially evident in medium to large datasets such as PubMed, Amp-Comp., DD, etc.
> >
> > *Table (3): The average elapsed time per epoch of the models on Node classiﬁcation datasets.*
> > |  Models  | Cora | CiteSeer | PubMed | WiKiCS | Amp-Comp. | Amp-Photo | Coauthor-CS |
> > | :--------: | ----: | ---------: | ---------: | ---------: | ---------: | ---------: | ---------: |
> > |  **GALOPA(cube)**  | 1.53s |    2.18s | 74.58s | 25.66s |    34.73s | 11.60s | 71.17s |
> > |   **GALOPA(linear)**   | 0.29s | 0.80s | 10.20s | 3.14s | 5.24s | 1.66s | 32.03s |
> >
> > *Table (4): The average elapsed time per epoch of the models on Graph classiﬁcation datasets.*
> >
> > |  Models  | PROTEINS | DD | MUTAG | NCI1 | COLLAB | IMDB-B |
> > | :--------: | ----: | ---------: | ---------: | ---------: | ---------: | ---------: |
> > |  **GALOPA(cube)**  |    7.05s | 300.50s | 1.21s | 18.15s | 74.64s |  3.59s |
> > |   **GALOPA(linear)**   |    3.36s |  20.07s | 0.47s | 11.75s | 21.46s | 2.76s |
> >
> > [1] Linear time sinkhorn divergences using positive features. NeurIPS2020.

---

> > ### Author Response · Authors · 2023-08-21
> > **Response (3/3) to Reviewer YJ1t**
> >
> > > **Q9-3. It might be more convincing if the new experiments on ogbn-arxiv can be timed for both the proposed method and related baselines.**
> >
> > Thanks to the suggestion of the reviewer, we record in the following table the average elapsed time per epoch taken to pre-train on the ogbn-arxiv with the algorithm GALOPA(cube), the variant algorithm GALOPA(linear), and the baseline BGRL (with linear complexity), and the fine-tuning accuracies obtained on the supervised algorithms.
> > All the experiments are conducted and runtimes are recorded on the *identical hardware environment* as described in the paper.
> > Note that for GALOPA(cube) and GALOPA(linear) we first partition the graph and compute the plan between subgraphs, where the average size of each subgraph is ~5000 nodes.
> >
> > The results of our experiments on the ogbn-arxiv dataset, presented in the following table, showcase the substantial reduction in running time achieved through the implementation of the complexity reduction approach. Importantly, this efficiency enhancement is coupled with comparable performance to the original model.
> >
> > *Table (5): The average elapsed time per epoch for training the models and the fine-tuning accuracies on ogbn-arxiv.*
> > |  Models  | Time | Test Accuracy |
> > | :--------: | :---: | :--------: |
> > |  **GALOPA(cube)**  | 60.4s | 0.544±0.18 |
> > |   **GALOPA(linear)**   | 2.79s | 0.541±0.29 |
> > | **BGRL** | 1.02s | 0.535±0.19 |
> >
> > > **Q10. Do you in practice use the gradient from** $\nabla_{Z_i}\pi^{(T)}$ **to update the parameters in GNN** $f()$**? If yes, even we know** $\pi^{(T)} \rightarrow \pi^*$**, how would you show the gradient** $\nabla_{Z_i} \pi^{(T)} \stackrel{T \rightarrow \infty}{\longrightarrow} \nabla_{Z_i} \pi^*$**?**
> >
> > We extend our gratitude to the reviewers for providing us with the opportunity to address this query. We use the Sinkhorn-Knopp algorithm for optimization and compute the gradient $\nabla_{Z_i}\pi^{(T)}$ using backpropagation to update the parameters of GNNs in the optimization progress.
> >
> > It's noteworthy that extensive research has been conducted concerning the convergence properties of this differential mechanism. Recent advancements in this domain are highlighted in the paper [2], wherein theoretical proofs have been established. Theorem 3.3 of [2] imples that $\pi^{(T)}$ is continuously differentiable for all $T$ and the sequence of derivatives $\nabla_{Z_i}\pi^{(T)}$ **converges** at a linear rate. In particular, for all $Z_i$, $\nabla_{Z_i}\pi^{(T)} \stackrel{T \rightarrow \infty}{\longrightarrow} \nabla_{Z_i}\pi^*$, where $Z_i$ is the variable of the cost matrix $\mathbf{C}(Z_i)$ and it corresponds to $\theta$ in the original text of [2]. $\mathbf{P}$ in [2] refers to the transport plan $\pi$.
> >
> > [1] Linear time sinkhorn divergences using positive features. NeurIPS2020.
> >
> > [2] The derivatives of sinkhorn–knopp converge. SIAM Journal on Optimization2023.
> >
> > **We really appreciate your valuable comments which help improve the quality of our manuscript, and we will add these comparisons and results to the final version.**

---

> ### Comment · Reviewer_YJ1t · 2023-08-22
>
> Many thanks for the new response. I would like to raise my ratings from 5 to 6 and encourage the authors to incorporate the discussion into the next revision.

---

> > ### Author Response · Authors · 2023-08-22
> > **Thank you!**
> >
> > Thank you very much for the update and your helpful suggestion, we'll add them to our next revision.

---

### Author Rebuttal · Authors · 2023-08-09

We sincerely thank all the reviewers for their insightful feedback.

Here we respond to common/main concerns raised by reviewers.

> **GQ1. The algorithms to compute the FGW (Eq. (6)) between graphs and the OT distance (Eq. (7)) between node embeddings. (zpGL, bARL)**

We describe the optimization algorithms for both of these terms below, which were previously mentioned in Appendix B of the original submission.

To solve the FGW problem of Eq. (6), we optimize the transport plan with the conditional gradient (CG) solver. The conditional gradient algorithm [1] consists in solving a linearization $\langle \mathbf{X}, \nabla_\pi \rangle$ at each iteration $r$. It can be solved by gradient descent with a direction $\mathbf{X}^{(r)} - \pi^{(r)}$, followed by a line search for the optimal step. The detail of the algorithm is summarized in Algorithm 1 in the Appendix.

To solve the Wassersttein problem of Eq. (7), in the paper we use the Sinkhorn-Knopp algorithm [2] to iteratively approximate the optimal solution $\pi_z^*$. Specifically, Sinkhorn-Knopp algorithm add an additional entropy regularizer and perform a scheme of alternating Sinkhorn projections: $\pi^{(0)}=\exp(-\boldsymbol{J}(\boldsymbol{Z}_1, \boldsymbol{Z}_2)/\lambda)$ and $\pi^{(t+1)} = \mathcal{S} (\mathcal{T}(\pi^{(t)}))$, where $t$ denotes the number of iterations, $\lambda$ weights the regularization, $\mathcal{S}(\pi)=\pi \oslash(\mathbf{1} \mathbf{1}^{\top} \pi) \odot(\mathbf{1} \boldsymbol{b}^{\top})$ and $\mathcal{T}(\pi)=\pi \oslash(\pi \mathbf{1} \mathbf{1}^{\top}) \odot(\boldsymbol{a} \mathbf{1}^{\top})$, $\odot$ denotes the Hadamard product and $\oslash$ denotes element-wise division. As shown by [2], in the limit this scheme converges to a minimizer $\pi^{(t)} \stackrel{t \rightarrow \infty}{\longrightarrow} \pi^*$.

[1] Revisiting frank-wolfe: Projection-free sparse convex optimization. ICML2013

[2] Sinkhorn distances: Lightspeed computation of optimal transport. NeurIPS2013



> **GQ2. The analysis and reducing of computational complexity. (YJ1t, bARL)**

**Analysis of computational complexity:** We provided a time complexity analysis in the appendix of our submission. We describe it in more detail below plus some additional comparisons with baseline algorithms: The time complexity of the GALOPA comes mainly from the optimization of Eqs. (6) and (7). For Eq. (6) containing Fused Gromov Wasserstein term, we use conditional gradient (CG) solver for optimization, which requires the computation of a gradient with near-cubic $\mathcal{O}(n^3)$ time complexity at each iteration, where $n$ denotes the size of graph, i.e., the number of nodes. For Eq. (7) with the Wasserstein term, we can use Sinkhorn-Knopp algorithm for time efﬁcient with near-square $\mathcal{O}(n^2)$ complexity.

**Reducing complexity**: To reduce the time complexity, we utilize the properties of the proposed model and/or the scaling optimal transport techniques that can reduce the time complexity from $\color{red}{\mathcal{O}(n^3)}$ to $\color{red}{\mathcal{O}(n^2)}$ or even to $\color{red}{\mathcal{O}(n)}$, we provide 4 ways to do this below:

1) Unlike general OT settings, where the two graphs are typically quite different and the matching relationship between them is completely unknown, the **difference between the original and augmented graphs** in GALOPA **is quite small** and the **matching relations for subgraph component** except with difference part (i.e., complementary set of difference part) **is known**. This means that we can utilize the *matching prior* to reduce the computational cost. Hence, we can split the difference part with its neighborhood from the two graphs and compute the optimal transport plan only for that part. Since the percentage of that part is very small, it can greatly reduce the time complexity. For example, with the perturbation rate is set to 1%, the time complexity of two million-sized graphs ($(10^{6})^3$ or $(10^{6})^2$) is directly reduced by 6 (or 4) orders of magnitude to $(10^6)^2$ or $(10^{6})^{1.3}$.

2) According to the observation in Section 6, we can avoid the cubic complexity $\mathcal{O}(n^3)$ of optimizing GW by **using only the node attributes** for computing the optimal plan in graph space, while retaining similar performance with near-square time complexity $\mathcal{O}(n^2)$.

3) Alternatively, we can reduce the computational cost by utilizing sparsity [1] or graph partitioning [2, 3]. In particular, we can employ the most recent work on linear optimal transport [4, 5], which compute FGW term and/or Wasserstein term in **linear time** $\mathcal{O}(n)$.

4) We have the option to combine the aforementioned methods. For instance, by merging insights from point 1, a significant portion of subgraph pairs acquired via graph partitioning methods in point 3 turns out to be **identical**. This realization can further pare down the complexity of graph partitioning methods.

[1] Efficient approximation of gromov-wasserstein distance using importance sparsification. JCGS2023

[2] Scalable gromov-wasserstein learning for graph partitioning and matching. NeurIPS2019

[3] Quantized gromov-wasserstein. PKDD2021

[4] Computing wasserstein-p distance between images with linear cost. CVPR2022

[5] On a linear fused gromov-wasserstein distance for graph structured data. PR2023

**Complexity of GCL:** We further analysis the time complexity of the contrastive learning algorithm. The contrastive loss (e.g., InfoNCE loss) computes all-pair distance for nodes as negative pairs which induces quadratic time complexity $\mathcal{O}(n^2)$ with respect to the graph size. Thus, general graph contrastive learning algorithms (e.g., GRACE, MVGRL, GraphCL, etc.) have quadratic time complexity $\mathcal{O}(n^2)$ with respect to graph size. BGRL introduced loss that does not require negative pairs with linear computation cost $\mathcal{O}(n)$.

---

### Decision · Program_Chairs · 2023-09-21

**Decision:**

Accept (poster)

**Comment:**

Graphs are notoriously hard to learn as their connections are arbitrary and the degrees of freedom are uncontrolled. With this, the paper presents a novel approach to the important task of self-supervised learning, a subject that attracts interest in the community. Although the work still have some drawbacks (e.g., runtime) that might influence the assimilation of these techniques, the novelty and possible applicability of the approach are of interest to the NeurIPS community.